# HIV-1 Nef Protein Affects Cytokine and Extracellular Vesicles Production in the GEN2.2 Plasmacytoid Dendritic Cell Line

**DOI:** 10.3390/v14010074

**Published:** 2021-12-31

**Authors:** Alessandra Aiello, Flavia Giannessi, Zulema Antonia Percario, Katia Fecchi, Claudia Arenaccio, Stefano Leone, Maria Carollo, Elisabetta D’Aversa, Laurence Chaperot, Roberto Gambari, Massimo Sargiacomo, Elisabetta Affabris

**Affiliations:** 1Department of Science, University Roma Tre, 00146 Rome, Italy; alessandra.aiello@uniroma3.it (A.A.); flavia.giannessi@uniroma3.it (F.G.); zulema.percario@uniroma3.it (Z.A.P.); claudia.arenaccio@uniroma3.it (C.A.); stefano.leone@uniroma3.it (S.L.); 2Center for Gender-Specific Medicine, Istituto Superiore di Sanità, 00161 Rome, Italy; katia.fecchi@iss.it; 3Core Facilities Technical Scientific Service—Next Generation Sequencing (NGS), Istituto Superiore di Sanità, 00161 Rome, Italy; maria.carollo@iss.it; 4Department of Life Sciences and Biotechnology, Biochemistry and Molecular Biology Section, University of Ferrara, 44121 Ferrara, Italy; elisabetta.daversa@unife.it (E.D.); gam@unife.it (R.G.); 5Immunobiology and Immunotherapy in Chronic Diseases, Institute for Advanced Biosciences, Inserm U 1209, CNRS UMR 5309, Université Grenoble Alpes, Etablissement Français du Sang Auvergne-Rhône-Alpes, 38000 Grenoble, France; laurence.chaperot@efs.sante.fr; 6National Center for Global Health, Istituto Superiore di Sanità, 00161 Rome, Italy; massimo.sargiacomo@iss.it

**Keywords:** plasmacytoid dendritic cells, HIV-1 Nef, cytokines, extracellular vesicles

## Abstract

Plasmacytoid dendritic cells (pDCs) are a unique dendritic cell subset specialized in type I interferon production, whose role in Human Immunodeficiency Virus (HIV) infection and pathogenesis is complex and not yet well defined. Considering the crucial role of the accessory protein Nef in HIV pathogenicity, possible alterations in intracellular signalling and extracellular vesicle (EV) release induced by exogenous Nef on uninfected pDCs have been investigated. As an experimental model system, a human plasmacytoid dendritic cell line, GEN2.2, stimulated with a myristoylated recombinant Nef_SF2_ protein was employed. In GEN2.2 cells, Nef treatment induced the tyrosine phosphorylation of STAT-1 and STAT-2 and the production of a set of cytokines, chemokines and growth factors including IP-10, MIP-1β, MCP-1, IL-8, TNF-α and G-CSF. The released factors differed both in type and amount from those released by macrophages treated with the same viral protein. Moreover, Nef treatment slightly reduces the production of small EVs, and the protein was found associated with the small (size < 200 nm) but not the medium/large vesicles (size > 200 nm) collected from GEN2.2 cells. These results add new information on the interactions between this virulence factor and uninfected pDCs, and may provide the basis for further studies on the interactions of Nef protein with primary pDCs.

## 1. Introduction

Plasmacytoid dendritic cells (pDCs) are one of the two principal subsets of human dendritic cells (DCs) and represent a link between innate and adaptive immunity [1,2]. Although constituting only 0.2–0.8% of human blood cells, they have garnered interest because they are able to produce up to 1000-fold more type I interferon (IFN) (particularly IFN-α) than any other cell types [3]. Different studies have shown that pDCs are involved in advanced inflammatory response in several autoimmune diseases and infections, including Human Immunodeficiency Virus (HIV) [4,5,6]. According to what was observed in a SIV (Simian Immunodeficiency Virus)-macaque model, pDCs are the first predominant cell type to arrive to infected mucosal sites where the infection is generally acquired [7]. Although they do not represent one of the main reservoirs of the virus (such as macrophages or CD4^+^ T lymphocytes), they can be infected as they express CD4 receptor and the co-receptors CXCR4 and CCR5, the surface molecules that are targeted by the virus [8,9,10,11]. It has been reported that pDCs might contribute dichotomously to both chronic immune activation and immunosuppression [12,13].

Over the years, the accessory protein Nef has been identified as one of the major determinants of HIV pathogenicity [14]. HIV-1 Nef (27–34 kDa, according to the isolate type) is a myristoylated, cytoplasmic multifunctional protein, partially associated with the cell membrane, that acts as a molecular adaptor inside the cells, exerting its effects through specific protein–protein interaction motifs [15,16]. Among the multiple functions ascribed to Nef, the hijacking of cellular signalling pathways and membrane trafficking have garnered the interest of the scientific community. Nef regulation of cellular signalling and trafficking pathways strongly suggests that it could influence *per se* the cytokine/chemokine network, possibly contributing to chronic inflammation, as observed for the first time in HIV-infected macrophages by the Mario Stevenson laboratory [17]. Previous studies conducted in our lab also demonstrated that the recombinant myristoylated Nef protein (myrNef_SF2_) was rapidly internalized in primary monocyte-derived macrophages (MDMs) and triggered NF-κB, MAPKs (Mitogen-Activated Protein Kinase) and IRF-3 (Interferon Regulatory Factor 3) activation, inducing the production and release of a set of cytokines/chemokines (CCL2/MIP- 1α and CCL4/MIP-1β, but also IL-6, TNF-α, IL-1β and IFNβ) [18,19]. The latter, in turn, activated some signal transducers and activators of transcription (STAT) molecules in an autocrine and/or paracrine manner, in particular STAT-1, -2 and -3 [18,20,21,22].

Much evidence also points to the ability of Nef to exploit the vesicular trafficking machinery of the host as a “Trojan horse” to be transferred through extracellular vesicles (EVs) and nanotubes from one cell to another, thus escaping the immune system and exerting its effects on both infected and uninfected cells [23,24,25]. EVs, including exosomes (30−150 nm diameter), formed as intraluminal vesicles (ILVs) in multivesicular bodies (MVBs), and microvesicles, (150−1000 nm diameter) budding directly from the plasma membrane [26], are membrane-bound vesicles naturally released from most cell types and recognized as potent vehicles of cell-to-cell communication. Nef-containing EVs have been reported to induce T-cell apoptosis [24], to make resting CD4^+^ T lymphocytes competent for HIV expression and replication, to reactivate cells latently infected with HIV-1 [27,28,29,30], as well as to enhance the levels of cytokines and chemokines such as IL-2, IL-8, IL-6, RANTES and IL-17A [31]. Although Nef has been consistently reported to increase EV release [23,24,32] and to be itself secreted in EVs, it remains unclear which type of EV is primarily involved, since Nef has been detected in both exosomes [24,33,34] and microvesicles [35], according to the cell type. Moreover, both Nef and anti-Nef antibodies were detected in the serum of HIV-infected individuals [36,37], supporting the possible in vivo detection of extracellular Nef by uninfected cells.

The discovery of multiple mechanisms by which Nef can be transferred during infection has opened a new frontier in the study of the multifaceted role of this viral protein. Since the effects of the pathogenic accessory protein Nef on pDCs have not been fully characterized, in this study, we examined the alterations in intracellular signalling and in the release of EVs induced by the treatment of non-HIV infected pDCs with myrNef. In particular, we used the human pDC cell line GEN2.2 as an experimental model system, demonstrating that myrNef treatment of these cells induced the release of a set of cytokines/chemokines which, in turn, activated STAT-1/2 proteins and influenced the gene expression program by inducing STAT1, IRF-1 and ISG15 expression. The produced set of cytokines/chemokines differed with respect to the one released by myrNef-treated differentiated human monocytic THP-1 cells. We also observed that myrNef treatment did not increase the EV release of GEN2.2 cells, and the protein was found to be associated with small (size < 200 nm) vesicles produced by the pDC cell line.

## 2. Materials and Methods

### 2.1. Cell Isolation and Culture

Peripheral Blood Mononuclear Cells (PBMCs) were isolated from buffy coats obtained from healthy donors at Centro Trasfusionale-Cattedra di Ematologia, Università degli Studi “La Sapienza” Rome. No ethical approval from University La Sapienza or Roma Tre ethics committees nor formal or verbal informed consent from blood donors were necessary to use buffy coats as sources of cells. PBMCs were isolated with Lympholyte-H (Cedarlane Laboratories Ltd., Burlington, ON, Canada) density gradient centrifugation and maintained in RPMI 1640 medium (Sigma-Aldrich, Milan, Italy) supplemented with 2 mM L-glutamine (Gibco, Amarillo, TX, USA), 100 Units/mL penicillin, 100 µg/mL streptomycin (Sigma-Aldrich, Milan, Italy) and 10% fetal bovine serum (FBS) (cat. 10270106, Gibco, Amarillo, TX, USA), previously inactivated at 56 °C for 30 min. Circulating pDCs were isolated from PBMCs by positive selection using an immunomagnetic-based kit (BDCA-four cell isolation kit, Miltenyi Biotec, Bologna, Italy), according to the manufacturer’s recommendations. The purified pDCs were maintained in RPMI 1640 medium supplemented with 2 mM L-glutamine, 100 Units/mL penicillin, 100 µg/mL streptomycin, 25 mM Hepes and 10% heat-inactivated FBS. PBMCs depleted of monocytes (PBLs), PBLs depleted of pDCs (PBLs-pDCs) and PBMCs depleted of pDCs (PBMCs-pDCs) were isolated by negative selection, and the cells were resuspended in RPMI 1640 medium supplemented with 2 mM L-glutamine, 100 Units/mL penicillin, 100 µg/mL streptomycin and 10% heat-inactivated FBS.

Since human primary pDCs are present in a very low amount in blood (0.2–0.5% of PBMCs), to facilitate the biochemical analyses of cell signalling, a set of experiments were carried out using GEN2.2, a pDC cell line derived from a leukaemia patient [38], deposited within the CNCM (Collection Nationale de Cultures de Microorganismes, Pasteur Institute, Paris) on 24 September 2002, under the number I-2938. The pDC cell line was obtained through a signed Material Transfer Agreement (MTA). The proliferation of GEN2.2 cells is strictly dependent on the presence of a feeder layer made by the murine stromal cell line MS-5 (deposited within the DSMZ (German Collection of Microorganisms and Cell Cultures) under the No. ACC441). GEN2.2 cells were cultured in flasks precoated with a sub-confluent irradiated MS-5 monolayer in RPMI 1640 medium containing 1% glutamax (Gibco, Gibco, Amarillo, TX, USA, cat. 35050-038), 100 Units/mL penicillin, 100 µg/mL streptomycin, 1 mM sodium pyruvate (cat. ECM0542D, Euroclone, Milan, Italy), 1% nonessential amino acids (Euroclone, Milan, Italy, cat. ECB3054D) (hereafter referred to as complete medium) and 10% ultra-low endotoxin FBS (cat. S1860-500, Microtech, Naples, Italy), previously inactivated at 56 °C for 30 min. GEN2.2 cells were maintained in culture for no more than two months and, only the CD45^+^ non-adherent fraction, corresponding to GEN2.2 cells in the supernatant, was used for the experiments.

THP-1 cells (from American Type Culture Collection, or ATCC), derived from a human monocytic leukaemia were grown in RPMI 1640 medium supplemented with 2 mM L-glutamine, 100 Units/mL penicillin, 100 µg/mL streptomycin and 10% heat-inactivated ultra-low endotoxin FBS. For the experiments, THP-1 were seeded at 100.000 cells/cm^2^ and added with 35 nM of PMA (phorbol 12-myristate 13-acetate) (cat. P8139, Sigma-Aldrich, Milan, Italy) to adhere and differentiate, acquiring a macrophage-like phenotype which mimics, in many respects, primary human macrophages [39]. In particular, after a PMA treatment of 32 h, the medium was replaced with fresh medium supplemented with 20% FBS. Then, after one day of resting, differentiated THP-1 were used for the experiments.

All cells were maintained at 37 °C in an incubator with a 5% CO_2_ humidified atmosphere.

### 2.2. Recombinant Nef Protein Preparations and Reagents

Wild type recombinant myristoylated Nef protein and a mutant in the acidic cluster E^66^EEE^69^→AAAA present at N-terminal end (referred to as myrNef_SF2_w.t and myrNef_SF2_4EA, respectively) were generated from HIV-1 SF2 allele in the laboratory of Dr. Matthias Geyer at Max-Plank-Institut fur molekulare Physiologie, Dortmund, Germany, as previously reported [40]. Briefly, these proteins were obtained by co-transformation of an *E. coli* bacterial strain with two plasmidic expression vectors containing a codon-optimized Nef or a N-myristoyl-transferase coding sequence, respectively, and purified as C-terminal hexahistidine-tagged fusion proteins. The myristoylation of recombinant Nef proteins was verified by mass spectrometry at Dr. Geyer’s laboratory. All Nef preparations were analysed for the presence of endotoxin as a contaminant using the chromogenic Limulus amebocyte lysate assay (LAL-test) (Biowhittaker, Walkersville, MD, USA) and, if required, purified using the EndoTrap^®^ red Endotoxin Removal Kit (Cambrex Bio Science Inc., Walkersville, MD, USA). To avoid possible signalling effects due to residual undetectable lipopolysaccharide (LPS) traces in Nef preparations, we performed some experiments in the presence of 1 µg/mL of polymyxin B (Sigma-Aldrich, Milan, Italy), a cationic antibiotic that binds to the lipid A portion of bacterial LPS. In our hands, this polymyxin B treatment blocked the signalling activity of up to 100 endotoxin units (EU)/mL LPS without inducing any differences in the signalling events analysed. For this reason, all the experiments described below were conducted in the absence of polymyxin B. CpG class A (cat. ODN2216, Miltenyi Biotec, Bologna, Italy) was used as a positive control for the innate activation of immune cells, such as human PBMCs and pDCs.

For the Interferon (IFN) treatments of cells, human recombinant rIFN-β (Ares-Serono, Geneva, Switzerland), human rIFN–γ (cat. #300-02; EC Ltd., PeproTech, London, UK) and human rIFN-λ1/λ2 (hereafter referred to as IFN-λ), generously gifted by Dr. Eliana Coccia (Department of Infectious Disease, Istituto Superiore di Sanità, Rome, Italy), were used.

### 2.3. Flow Cytometry Analysis

The purity of the cells isolated from the peripheral blood of healthy donors was assessed by flow cytometry (FC) analysis. For surface staining, cells (10^5^) were resuspended in 50 µL of phosphate buffered saline (PBS) containing 2% FBS and incubated in the dark for 30 min at 4 °C with the corresponding mixture of antibodies (see Appendix A). As a control, we used isotype-matched antibodies labelled with the appropriate fluorochrome. After incubation, cells were washed, fixed with 2% paraformaldehyde (PFA) (Sigma-Aldrich, Milan, Italy) for 15 min on ice and, finally, left in 1% PFA until the observation with the cytofluorimeter CytoFLEX (Beckman Coulter, Brea, CA, USA). Since CD123 is the specific cell surface marker for plasmacytoid dendritic cells, the purity of these cell types was assessed by means of anti-CD123 monoclonal antibody labelling. The purity of the populations of PBMCs depleted of pDCs and of PBMCs depleted of monocytes (PBLs) was measured by evaluating, respectively, the percentage of CD123 and CD14 positive cells. Cell populations whose purity was below 95% were discarded.

The purity of GEN2.2 cells recovered from the co-culture with the MS-5 monolayer was assessed by analysing the expression of different markers. To this end, the following antibodies were used: fluorescein-5-isothiocyanate (FITC)-conjugated anti-HLA-DR (clone HI43, cat. 21819983), allophycocyanin (APC)-conjugated anti-CD44 (clone MEM-85, cat. 21270446), phycoerythrin (PE)-conjugated anti-CD123 (clone AC145, cat. 130-113-326, Miltenyi Biotec, Bergisch Gladbach, Germany), APC-conjugated anti-CD11c (clone BU15, cat. 21487116), FITC-conjugated anti-CD29 (clone MEM-101A, cat. 21270293) and FITC-conjugated anti-CD45 (clone MEM-233, cat. 21270453), FITC-conjugated anti-CD4 (clone MEM-241, cat. 21270043), FITC-conjugated anti-CD86 (clone BU63, cat. 21480863) and FITC-conjugated anti-CD80 (clone MEM-233, cat. 21270803) (all generously gifted by ImmunoTools GmbH, Friesoythe, Germany). For surface staining, GEN2.2 cells (1 × 10^6^) were processed as reported above. As a control, the autofluorescence of the cells was used.

### 2.4. Bodipy FL C16 Reconstitution and Cell Labelling

The quantification of EVs released by GEN2.2 cells was performed using the labelling protocol developed by Sargiacomo and colleagues [41]. This protocol was based on cell treatment with the commercially available BODIPY FL C16 fatty acid (4,4-difluoro-5,7- dimethyl-4-bora-3a,4a-diaza-s-indacene-3-hexadecanoic acid) (Life Technologies, Monza, Italy), hereafter indicated as Bodipy C16, a fluorescent lipid that labels the cells, ultimately producing fluorescent vesicles. Briefly, the fluorescent lipid was resuspended in methanol at 1 mM final concentration and stored at −20 °C in aliquots of 150 µL. Before use, each aliquot was dried under nitrogen gas at room temperature, resuspended with 30 µL of 20 mM KOH to avoid the formation of micelles and to promote its solubilisation, heated for 10 min at 60 °C and finally resuspended in 70 µL of PBS containing 2% of bovine serum albumin (BSA).

For pulse-chase studies, 1 × 10^7^ GEN2.2 cells were metabolically labelled with Bodipy C16 at different times and concentrations, as reported in the text. Importantly, to favour the uptake of the fluorescent probe, the treatments were performed using complete medium supplemented with only 0.3% FBS. Afterwards, cells were washed with 1× PBS to remove lipid excess, and complete culture medium supplemented with 10% FBS was added. The fluorescence intensity of GEN2.2 cells was evaluated by flow cytometry analysis and reported in terms of mean fluorescence intensity (MFI), and then observed by confocal microscopy.

For the isolation and quantification of fluorescent EVs, 1 × 10^7^ GEN2.2 cells were seeded in 75 cm^2^ flasks and incubated for 2 h at 37 °C, with 3.5 µM of Bodipy C16 in 5 mL of medium supplemented with 0.3% FBS. Then, cells were washed with 1× PBS and resuspended in 12 mL of complete culture medium supplemented with 10% FBS, containing or not myrNef_SF2_w.t. The FBS added to the medium was previously ultracentrifuged overnight for 18 h at 100,000× *g* in a SW41 Ti rotor (Beckman Coulter, Brea, CA, USA), to remove the EVs normally present in serum.

### 2.5. Extracellular Vesicle Purification

EVs were isolated from identical volumes (12 mL) of cell conditioned and non-conditioned control media, which were harvested after 20 h and processed following the already described methods for EV purification [42]. Briefly, cell cultures or culture medium, used as a control, were centrifuged at 290× *g* for 7 min to remove cells and then at 2000× *g* for 20 min to eliminate cell debris. Subsequently, supernatants underwent differential centrifugations consisting of a first ultracentrifugation at 15,000× *g* for 20 min to isolate large/medium EVs (hereafter referred to as microvesicles). To isolate small EVs (referred to as exosomes), supernatants were then harvested and ultracentrifuged at 100,000× *g* for 3 h. The pelleted vesicles were left for 20 min on ice and then resuspended in 12 mL of 1× PBS and ultracentrifuged again at 100,000× *g* for 3 h. All ultracentrifugation steps were performed at 4 °C using a SW41 Ti rotor (Beckman Coulter, Brea, CA, USA). Isolated exosomes and microvesicles were finally resuspended in 100–200 µL of PBS with protease and phosphatase inhibitors (1 mM sodium orthovanadate, 20 mM sodium fluoride, 1 µg/mL leupeptin and pepstatin A, 2 μg/mL aprotinin and 1 mM phenylmethylsulfonyl fluoride (PMSF)) and stored at 4 °C until counting by flow cytometry and further analyses.

### 2.6. Quantification of Vesicles by Flow Cytometry

Flow cytometry of Bodipy-labelled EVs was performed on a Gallios Flow Cytometer (Beckman Coulter, Brea, CA, USA) at Istituto Superiore di Sanità (Via Regina Elena 299, Rome, Italy) under the supervision of Dr. Katia Fecchi and Dr. Maria Carollo, using an optimized procedure as previously described [41]. Briefly, 5 μL of fluorescent exosomes or microvesicles were mixed with 20 μL of Flow-Count Fluorospheres with a size of 100 nm (Beckman Coulter, Brea, CA, USA), which were used as an internal reference standard, and further diluted with 1× PBS to a final volume of 200 μL. The instrument was set up using control 100–500 nm fluorescent beads in order to identify the right gate corresponding to exosomes (size below 200 nm). FC analysis was performed by plotting fluorescence at 525/40 nm (FL1) versus log scale side scatter (SS area). The instrument was set at high flux and the analysis was stopped at 2000 Flow-Count Fluorospheres events. Fluorescent EVs’ total number was established according to the formula:EVs = (((y × a)/b)/c) × d,
where y = events counted at 2000 counting beads; a = number of counting beads in the sample; b = number of counting beads registered (2000); c = volume of sample analysed and d = total volume of exosome preparation. The total number of exosomes and microvesicles obtained was then normalized against the number of cells counted after 20 h of treatment. Kaluza Software v. 2.0 (Beckman Coulter, Brea, CA, USA) was used for FC analysis.

### 2.7. Western Blot Assay

Western blot analyses on cell lysates were performed by washing cells twice with ice-cold PBS (pH 7.4) and lysing them for 30 min on ice with lysis buffer (50 mM Tris pH 7.4, 150 mM NaCl, 0,25% sodium deoxycholate, 1 mM EDTA, 1 mM EGTA, 1% Triton X-100, 0.5% non-ionic detergent IGEPAL CA-630 (Sigma-Aldrich, Milan, Italy), 1 mM sodium orthovanadate, 20 mM sodium fluoride, 1 µg/mL leupeptin and pepstatin A, 2 μg/mL aprotinin and 1 mM phenylmethylsulfonyl fluoride (PMSF)). Whole-cell lysates were centrifuged at 6000× *g* for 10 min at 4 °C. The protein concentration of cell extracts was determined by the Lowry protein quantification assay. Aliquots of cell extracts containing 30 to 50 µg of total proteins were resolved by 6 to 13.5% sodium dodecyl sulphate-polyacrylamide gel electrophoresis (SDS-PAGE) and transferred by electroblotting on 0.45 µm pore size nitrocellulose membranes (AmershamTM, GE Healthcare Life Science, Milan, Italy) overnight at 35 V using a Bio-Rad Mini Trans-Blot Cell. For Western blot analysis of EVs, they were lysed by repeated freezing and thawing and then processed as described for cell lysates.

For the immunoassays, membranes were blocked in 3% bovine serum albumin (BSA) fraction V (Biofroxx, Einhausen, Germany) in TTBS/EDTA (10 mM Tris pH 7.4, 100 mM NaCl and 1 mM EDTA, 0.1% Tween 20) for 30 min at room temperature (RT) and then incubated for 1 h at RT or overnight at 4 °C with specific primary antibodies diluted in 1% BSA/TTBS-EDTA. The antibodies used in immunoblottings were the following: rabbit polyclonal anti-phosphotyrosine (Y701) STAT1 (Cell Signalling, Beverly, MA, USA, cat. #9171), mouse monoclonal anti-STAT1 (Transduction Laboratories, Milan, Italy, cat. #G16920), rabbit polyclonal anti-phosphotyrosine (Y689) STAT2 (Upstate Biotech/Millipore, Burlington, MA, USA, cat. #07-224), rabbit polyclonal anti-STAT2 (Santa Cruz Biotechnology, Dallas, TX, USA, cat. #sc-476), mouse monoclonal anti-ISG15 (Santa Cruz Biotechnology, Dallas, TX, USA, cat. #sc-166755), rabbit polyclonal anti-IRF-1 (Santa Cruz Biotechnology, Dallas, TX, USA, cat. #sc-497), rabbit polyclonal anti-α-actin (Sigma-Aldrich, Milan, Italy, cat. #A2066), rabbit polyclonal anti-Lamin A (Abcam, Cambridge, UK, cat. #ab26300), mouse monoclonal anti-TSG101 (Genetex, Irvine, CA, USA, cat. #GTX70255), rabbit polyclonal anti-Alix (Novus Biologicals, Milan, Italy, cat. #NBP1-90201), rabbit polyclonal anti-Hsp90 (Santa Cruz Biotechnology, Dallas, TX, USA, cat. #sc-7947), mouse monoclonal anti-CD81 (Santa Cruz Biotechnology, Dallas, TX, USA, cat. #sc-166029), mouse monoclonal anti-Flotillin-1 (BD Biosciences, Milan, Italy, cat. #610821), mouse monoclonal anti-COX4 (Santa Cruz Biotechnology, Dallas, TX, USA, cat. #sc-376731) and mouse anti-Nef ARP3026 (obtained from the NIH AIDS Research and Reference Reagent Program). The immune complexes were detected by incubating membranes for 1 h at RT with horseradish peroxidase-conjugated goat anti-rabbit (Merk Millipore, Burlington, MA, USA, cat. #AP307P) or goat anti-mouse (Enzo Life Technologies, Farmingdale, NY, USA, cat. #ADI-SAB-100-J) antibodies, followed by enhanced chemiluminescence reaction (ECL Fast Pico; Immunological Sciences, Rome, Italy). To reprobe membranes with antibodies having different specificities, nitrocellulose membranes were stripped for 5 min at RT with Restore Western Blot Stripping Buffer (Thermo Scientific™ Pierce™ Protein Biology, Rockford, IL, USA) and then extensively washed with TTBS/EDTA.

The ChemiDoc XRS (Bio-Rad, Hercules, CA, USA) instrument and the Image Lab software (Bio-Rad) were used to reveal the chemiluminescence signal. For loading control, α-actin levels were quantified by using a rabbit polyclonal anti-α-actin antibody (Sigma-Aldrich, Milan, Italy, cat. #A2066). Densitometric analyses were performed, using the freeware Image J software (NIH), by quantifying the band intensity of the protein of interest with respect to the relative loading control band (i.e., actin) intensity. Fold changes of each analysed protein were calculated by dividing the values obtained in treated conditions by those of the corresponding controls, and were reported in histograms as means ± S.D. of at least three independent experiments.

### 2.8. Nuclear and Cytoplasmic Extract Preparation

GEN2.2 (4 × 10^6^ cells) were treated with myrNef_SF2_w.t or IFNs for 20 h, harvested and washed twice in ice-cold PBS buffer by centrifuging at 1200 rpm for 3 min at 4 °C. Cell pellets were lysed with 200 µL of hypotonic buffer (10 mM Hepes pH 7.8, 10 mM KCl, 1 mM MgCl2, 0.1 mM EGTA, 0.5 mM EDTA, 5% glycerol, 1 mM PMSF, 1 μg/mL pepstatin A, 2 μg/mL aprotinin, 1 μg/mL leupeptin, 1 mM Na3VO4 and 20 mM NaF) and incubated on ice for 15 min. Afterwards, 0.58% IGEPAL CA-630 (Sigma-Aldrich, Milan, Italy) was added, incubated on ice for 2 min and then centrifuged at 14,000 rpm for 5 min. Supernatants corresponding to the cytoplasmic fraction were harvested, whereas 60 µL of hypertonic buffer (50 mM Hepes, pH 7.8; 400 mM NaCl; 1 mM MgCl2; 1 mM EGTA; 1 mM EDTA; 10% glycerol; 1 mM PMSF; 1 µg/mL pepstatin A; 2 µg/mL aprotinin; 1 µg/mL leupeptin; 1 mM Na3VO4 and 20 mM NaF) were added to the nuclear pellets, then incubated on ice for 40 min and centrifuged at 14,000 rpm for 10 min. Supernatants corresponding to the nuclear fraction were finally harvested. Protein concentrations of both cytoplasmic and nuclear fractions were determined by the Lowry protein quantification assay. All samples were processed and analysed by Western blot, as previously described.

### 2.9. Confocal Microscopy

To evaluate the internalization of Nef protein by Confocal Laser Scanner Microscopy analysis, primary human pDCs and GEN2.2 cells were seeded at 10^5^ cells/200 µL and 0.2 × 10^6^ cells/150 µL, respectively, in complete 10% FBS medium in 96-well plates and treated with 300 ng/mL of myrNef_SF2_w.t-Alexa488 or myrNef_SF2_4EA-Alexa488, which were labelled using AlexaFluor488 Microscale Protein Labelling Kit (Molecular Probes/Invitrogen, Monza, Italy) following the manufacturer’s recommendations. Cells were harvested at indicated times, washed once in 1× PBS, placed on the microscope slide and left to air dry. Subsequently, they were fixed with 4% PFA for 15 min on ice and then washed three times with PBS. Finally, coverslips were mounted using Vectashield antifade mounting medium (Vectashield H-1000; Vector Laboratories Inc., Burlingame, CA, USA) diluted at 80% in PBS to prepare samples for confocal microscopy observation. Plasma membrane counterstaining was performed by treating primary pDCs for 5 min with PKH26-GL, using the PKH26 Red Fluorescent Cell Linker Kit for General Cell Membrane Labeling (Sigma-Aldrich, Milan, Italy) following the manufacturer’s recommendations. Nuclei of GEN2.2 cells were stained with 3 µg/mL DAPI (4′, 6′-diamidino-2-phenylindole) (Sigma-Aldrich, Milan, Italy) that was directly added to the mounting medium.

In order to assess IRF-7 increase, primary pDCs were seeded at 10^5^ cells/200 µL in complete 10% FBS medium in 96-well plates and treated with myrNef_SF2_w.t (300 ng/mL) or CpG A (3 μg/mL). Primary pDCs were fixed with 4% PFA for 15 min on ice, then washed three times with PBS and permeabilized with 0.1% Triton X-100 in PBS for 10 min on ice. Afterwards, the specimens were incubated for 30 min in the dark at RT with 1% BSA in PBS containing far-red fluorescent dye RedDotTM2 to stain nuclei (Biotium, Inc. Hayward, CA, USA), washed and then incubated in the dark for 1 h at RT with the following antibodies: rabbit anti-IRF-7 antibody (Santa Cruz Biotechnology, Dalls, TX, USA, cat. #sc-9083), diluted 1:50 in 0.1% BSA in PBS, and AlexaFluor546-conjugated anti-rabbit (Life Technologies, Monza, Italy, cat. #A11010) as a secondary antibody, diluted 1:200 in 0.1% BSA in PBS. Finally, the specimens were washed four times in PBS and prepared for confocal microscopy observation, as previously described.

For pulse-chase studies, 3 × 10^5^ GEN2.2 cells were seeded in 48-well plates and metabolically labelled with Bodipy C16 according to the concentrations and interval of times reported. Cells were then washed twice with 1× PBS, placed on a microscope slide and fixed as reported above. Finally, samples were mounted with Vectashield antifade mounting medium containing DAPI for nucleus staining.

All samples were stored protected from the light at –20 °C until the observation. Images were acquired with Leica TCS SP5 confocal microscope and processed with LAS AF software (version 1.6.3, Leica Microsystems CMS GmbH). Objective 63.0X. Lasers activated: Argon laser at 488 nm to visualize myrNef_SF2_-Alexa488 (green) and UV laser at 405 nm to observe nuclei stained with DAPI. Images were acquired activating single laser in sequential mode to prevent fluorescence overlay. Several fields were analysed for each condition and representative results are shown.

### 2.10. RNA Extraction and Quantitative RT-PCR Analysis

For RNA extraction, cells were seeded at 10^6^ cells/mL and treated for 6 h with 300 ng/mL of myrNef_SF2_w.t or with 3 μg/mL of CpG-A, as a positive control, or left untreated. After treatment, cells were washed with ice-cold PBS and centrifuged at 290× *g* for 10 min. Cell pellets were lysed in RLT lysis buffer containing β-mercaptoethanol (Qiagen Inc, Valencia, CA, USA), and then RNA was isolated using the High Pure RNA Isolation Kit from Qiagen, according to the manufacturer’s recommendations. The amount of RNA extracted was measured by means of Nanodrop spectrophotometer (Thermo Scientific, Wilmington, DE, USA). The retrotranscription was performed using 0.5–1 µg of mRNA and the Murine Leukemia Virus Reverse Transcriptase (Invitrogen, Life Technologies, Monza, Italy). According to the protocol, mRNA was incubated for 1.5 h at 37 °C with a mixture containing 1 μM oligo-dT12-18, 1 µM random primers, 0.5 mM deoxynucleotides triphosphates (dNTPs), 10 mM DTT, first Strand Buffer 5X (250 mM Tris-HCl pH 8.3, 375 mM KCl and 15 mM MgCl2), 0.04 U/μL of ribonuclease inhibitor RNasiOUT™ and, finally, 8 U/μL of retrotranscriptase. The obtained cDNA was then purified using the QIAquick PCR Purification Kit (Qiagen Inc, Valencia, CA, USA), following the manufacturer’s instructions.

Quantitative PCR assays to evaluate the expression of *mxA* gene were performed with SYBR Green I technology on the Light Cycler instrument (Roche Diagnostics GmbH). In particular, 2 µL of template cDNA were added in a final volume of 20 µL, containing a mix of forward and reverse primers (500 nM each one) specific for the analysed gene (synthesized at Eurofins MWG Operons), the Platinum Taq DNA enzyme Polymerase (Invitrogen Life Technologies, Monza, Italy) and SYBR Green I (Biowhittaker Molecular Applications, Rockland, ME, USA). In detail, primers used were the following: forward, 5′-ATCCTGGGATTTTGGGGCTT-’3 and reverse 5′-CCGCTTGTCGCTGGTGTCG-’3. The data shown were normalized using the 2^−∆Ct^ formula, where ∆Ct represents the difference between the amplification cycles of *mxA* gene and the amplification cycles of the housekeeping gene GAPDH (glyceraldehyde-3-phosphate-dehydrogenase), constitutively expressed in all cell types.

### 2.11. Bio-Plex Analysis

GEN2.2 cells were cultured at 10^6^ cells/mL in complete 10% FBS medium in 24-well plates, whereas THP-1/PMA cells were seeded at 100,000 cells/cm^2^ in a 6-well plate. Both cell types were stimulated with 300 ng/mL of myrNef_SF2_w.t or myrNef_SF2_4EA or left unstimulated. Supernatants were harvested after 4, 6 and 20 h, centrifuged at 290× *g* for 3 min to eliminate cells and then stored at –80 °C until cytokine measurement. In collaboration with Professor Roberto Gambari at University of Ferrara, supernatants were analysed in a Bio-Plex Pro Human Cytokine 27-Plex Immunoassay (Bio-Rad, Hercules, CA, USA) according to the manufacturer’s instructions. The multiplex allowed detection of the following cytokines: FGF basic, Eotaxin, G-CSF, GM-CSF, IFN-γ, IL-1β, IL-1ra, IL-2, IL-4, IL-5, IL-6, IL-7, IL-8, IL-9, IL-10, IL-12 (p70), IL-13, IL-15, IL-17A, IP-10, MCP-1 (MCAF), MIP-1α, MIP-1β, PDGF-BB, RANTES, TNF-α and VEGF in a single well. Briefly, an amount of 50 μL of cytokine standards or samples was incubated with 50 μL of anti-cytokine-conjugated magnetic beads in a 96-well plate for 30 min at room temperature with shaking. The plate was then washed three times with 100 μL of Bio-Plex Wash Buffer using the Bio-Plex Pro Wash Station (Bio-Rad, Hercules, CA, USA); 25 μL of diluted detection antibody were added to each well, and the plate was incubated for 30 min at room temperature with shaking. After three washes, 50 μL of streptavidin-phycoerythrin were added and the plate was incubated for 10 min at room temperature with shaking. Finally, the plate was washed three times, the beads were suspended in Bio-Plex Assay Buffer and the samples were read using the Bio-Rad 96-well plate reader. Data were analysed using the Bio-Plex Manager Software (Bio-Rad, Hercules, CA, USA).

### 2.12. Statistical Analysis

Differences were statistically evaluated using a two-tailed Student’s *t* test to calculate significant differences between two groups and one-way ANOVA and Tukey’s multiple comparisons to calculate significant differences between three or more groups. Data were analysed with GraphPad Prism 8 software. *p* values ≤ 0.05 were considered statistically significant. * *p* < 0.05, ** *p* < 0.01, *** *p* < 0.005.

## 3. Results

### 3.1. myrNef_SF2_ Induces the Tyrosine Phosphorylation of STAT1 in Human PBLs but Not in PBLs Depleted of pDCs, and Increases mxA Expression

Previous studies carried out on primary monocyte-derived macrophages (MDMs) showed that myrNef_SF2_ indirectly activated some STAT (Signal Transducers and Activators of Transcription) family members (i.e., STAT-1, -2 and -3) in an autocrine and/or paracrine manner by inducing in 2 h the production and secretion of a number of pro-inflammatory factors and IFN beta [18,20,21,22]. These findings prompted us to analyse the effect of the viral protein on other cell types present in the PBMC population by evaluating the tyrosine (Y701) phosphorylation of STAT1, a transcriptional factor usually activated in response to a wide range of cytokines, including IFNs. The experiments were initially carried out on PBLs, a population that includes mainly B and T lymphocytes, natural killer cells, myeloid dendritic cells and pDCs.

PBLs were isolated from PBMCs by negative selection removing CD14 positive cells (monocytes). The efficiency of the cell depletion and the purity of the recovered cells were determined by flow cytometry analyses (Appendix A).

To appropriately monitor and characterize possible effects on STAT1 activation, PBLs were treated for different time intervals with myrNef_SF2_w.t (i.e., 2, 4 or 6 h). As shown, myrNef_SF2_w.t induces the tyrosine (Y701) phosphorylation of STAT1 in PBLs, starting at 4 h, and the signal also persists at 6 h (Figure 1A,B), confirming what was previously observed in macrophages. To identify the responsive cell population, PBLs were depleted of T lymphocytes and then treated with the viral protein. As shown in Figure 1C,D, CD3^−^ cells, including B lymphocytes, natural killer and dendritic cells, still showed the phosphorylation of STAT1. Subsequently, PBLs were depleted of pDCs in order to evidence the role of this dendritic subset in the response. We observed that PBLs depleted of pDCs failed to respond to Nef stimulus (Figure 1E,F). This preliminary result suggested that pDCs could have a particular importance in the response of PBLs to the viral protein Nef.

Since pDCs are widely recognized as the main producers of type I IFN, we also asked whether Nef protein induced the expression of the IFN inducible gene *mxA* (myxovirus resistance protein A). The *mxA* protein was chosen because it is a key mediator of the antiviral response induced by IFNs against a wide variety of viruses. Moreover, its expression is strictly regulated by type I and III IFNs, requires functional activation of STAT1 and is not directly induced by viruses or other stimuli [43]. The experiment was carried out using total PBMCs and PBMCs depleted of pDCs (PBMCs-pDCs). Both cell types were treated for 6 h with myrNef_SF2_w.t (300 ng/mL) or with CpG A (1 µM), a TLR9 agonist in response to which pDCs synthesize high levels of IFN-α as a positive control. The results showed that Nef increased *mxA* expression in both PBMCs and PBMCs-pDCs, but a reduction in this increase was observed when PBMCs were depleted of pDCs (Figure 1G). This result suggests that Nef treatment increases *mxA* in pDCs, contributing to the higher response observed in PBMCs. Altogether, these data prompted us to address our work on this particular dendritic subset.

### 3.2. Nef Induces the Increase and Nuclear Translocation of IRF-7 in Primary pDCs

First, we evaluated the capability of these cells to internalize the recombinant protein. To this aim, primary pDCs were isolated from PBMCs by positive selection, using BDCA-4 conjugated microbeads and assayed for their purity by FC analysis (Figure 2A). Isolated pDCs were treated with 300 ng/mL of myrNef_SF2_-AlexaFluor488 for 24h. Confocal microscopy images showed that Nef protein was internalized by primary pDCs (Figure 2B). The observation of several fields (for a total of about 500 cells) revealed that approximately 30% of pDCs internalized the viral protein.

Since both type I (α/β) and type III (λ) IFN can regulate the expression of *mxA* gene, and their expression depends on a similar transcription model that requires the previous activation and nuclear translocation of specific IFN regulatory factors (IRFs), such as IRF-7 [44], we evaluated whether Nef treatment induced the activation and nuclear translocation of this factor in pDCs. To this aim, primary pDCs were treated with myrNef_SF2_w.t (300 ng/mL) for 6 and 20 h, and with CpG A for 20 h as a positive control. Afterwards, cells were harvested and labelled in order to observe IRF-7 by confocal microscopy (Figure 3). The images revealed that IRF-7 was increased and, although it was mainly localized in the cytoplasm, a partial nuclear translocation was detected after 20 h of Nef treatment. Moreover, a basal expression of IRF-7 in untreated cells was observed, in agreement with literature reporting that plasmacytoid dendritic cells constitutively express not only IRF-3, but also IRF-7 [45].

### 3.3. GEN2.2 Cell Line as a Model System for Studying the Effects Induced by HIV-1 Nef on pDCs

The promising results obtained in primary pDCs led us to better investigate the effects induced by Nef protein on this unique dendritic cell subset. To facilitate biochemical analyses of cell signalling, which are difficult to perform in rare and in vitro short living human primary pDCs, we decided to use a human pDC cell line called GEN2.2. This cell line shares most of the phenotypic and functional features of primary pDCs [38,46], thus it was chosen in order to have a more stable and reproducible system.

The immunophenotype of GEN2.2 cells was analysed by flow cytometry for the expression of different markers, known to be present on the surface of primary pDCs, to verify the purity of the cells recovered from the co-culture with MS-5 cell line (Appendix A). Independently from the time spent in culture, GEN2.2 cells, like human primary pDCs, expressed CD4, the main cellular receptor mediating HIV binding in pDCs, HLA-DR, CD123, CD44, CD29 and CD45. The latter is not expressed by MS-5 cells. As expected, GEN2.2 cells were negative for CD11c, a myeloid dendritic cell marker. Moreover, they expressed high levels of CD86, whereas CD80 was undetectable (Appendix A). GEN2.2 cells proliferate rapidly as a single cell suspension with both non-adherent and weakly adherent cells, but for the experiments, only the CD45^+^ non-adherent fraction of the culture was used. Then, the internalization of the recombinant Nef protein was evaluated by treating GEN2.2 cells with myrNef_SF2_w.t conjugated with AlexaFluor488 for different time points (Figure 4A). As shown by confocal images, myrNef_SF2_ was already taken up by the cells after 4 h, and its uptake was increased after 20 h without significant variations in the number of cells that internalized the protein. Importantly, the analysis of several fields (for a total of about 2000 cells) revealed that approximately 50% of GEN2.2 cells internalized the protein after 4 h, but with different efficiency among them. To further confirm the internalization, a Western blot analysis was performed (Figure 4B,C). To this end, GEN2.2 cells were treated with increasing concentrations of myrNef_SF2_w.t for 4 h. The extent of the protein inside the cellular extract correlated with Nef input. Remarkably, the viral protein was detectable in the extract starting from a treatment with 200 ng/mL. Moreover, we evaluated whether the viral protein induced the tyrosine (Y701) phosphorylation of STAT1. We observed that GEN2.2 cells treated with 300 ng/mL of myrNef_SF2_w.t responded more strongly, in addition to presenting a well-detectable amount of the protein inside the cells. Hence, the following experiments were performed using this protein concentration. Considering these results, we can infer that GEN2.2 cells are less sensitive to Nef treatment with respect to what was previously observed in primary macrophages. In particular, in primary macrophages, STAT1 tyrosine phosphorylation is induced by the release of cytokines and chemokines with lower concentrations of the viral protein (10-100 ng/mL) and earlier, i.e., after only 2 h of cell treatment with Nef [18,20]. Concurrently, the same analyses were also performed by treating GEN2.2 cells with the mutant myrNef_SF2_4EA, whose acidic cluster domain at amino acids (aa) 66 to 69 was inactivated by the substitution with four alanines. This Nef mutant is internalized by macrophages, but it is not able to induce the release of the STATs’ activating factors [18,19]. As shown by confocal images and confirmed by Western blot analysis, the mutant 4EA was also internalized by GEN2.2 cells (45.3% after 4 h and 55.8% after 24 h) without significant differences compared to wild type Nef (Figure 4A and lower panel of Figure 4B).

Altogether, these data contribute to validating GEN2.2 cell line as an appropriate experimental model system.

### 3.4. HIV-1 myrNef_SF2_w.t Protein Induces STAT-1 and -2 Tyrosine Phosphorylation and Increases the Expression of Proteins Transcriptionally Regulated by Their Activation

Specific STAT activation occurs in a very short time (i.e., minutes) after cell treatment with the activating factors through the JAK-STAT signal transduction pathway. Afterwards, specific inhibitors of the JAK/STAT signal transduction pathway gradually switches off STAT activation. Therefore, GEN2.2 cells were treated with myrNef_SF2_w.t or the mutant myrNef_SF2_4EA using two different kinetics: the shorter one (2–4 h) (Figure 5A) and the longer one (5–40 h) (Figure 5B). The mutant 4EA was used to verify whether the Nef acidic cluster domain was as important for this signalling as it is in macrophages [19].

As shown, myrNef_SF2_w.t, but not its mutant 4EA, increased the tyrosine phosphorylation of STAT1 (Y701) and STAT2 (Y689) starting from 3 h; the signal still persisted at 6 h, but returned at the basal level after 20 h of treatment (Figure 5A–D). Of note, wild type Nef also induced an increase in the levels of STAT1-α/β protein expression, which became clearly detectable 20 h after the treatment persisting up to 40 h (Figure 5B,D). Moreover, to assess whether Nef-induced STAT1 and STAT2 activation influenced the expression of other STAT1 and/or STAT2 regulated genes, the expression of IRF-1 and ISG15 was evaluated. These two genes are transcriptionally regulated by the GAF (STAT1 homodimer) and the ISGF3 (STAT1-STAT2-IRF9) transcription factor, respectively.

IRF-1 is a transcription factor of the IRF (Interferon Regulatory Factor) family that is transiently up-regulated by type I IFN and persistently up-regulated by type II IFN via the GAF factor. ISG15 is an ubiquitin-like modifier that is transcriptionally induced by type I and III IFNs, viral and bacterial infections. The latter exists as a 17-kDa precursor protein that is rapidly processed into its mature 15-kDa form via protease cleavage to expose a carboxy-terminal motif, which allows the covalent binding of ISG15 to target proteins by a three-step process referred to as ISGylation [47]. ISGylation of TSG101, a transmembrane protein belonging to the Endosomal Sorting Complex Required for Transport (ESCRT) involved in the exosome biogenesis, can inhibit exosome secretion [48]. ISG15 also exists as an unconjugated protein that can be released into the extracellular milieu via non-conventional secretion, including exosomes [49]. The unconjugated form of ISG15 is the one that was analysed in Figure 5. As shown, wild type Nef, but not the 4EA mutant, increased IRF-1 amount transiently (only after 6 h), whereas ISG15 production started to be induced at 6 h, increased about three-fold after 20 h and was still well detectable after 40 h (Figure 5B,D). The phosphorylation of both STAT1 and 2 and the kinetics of increase in IRF-1 and ISG15 are compatible with the induction of type I and/or III production by the Nef treated cells.

Interestingly, a further analysis of the nuclear and cytoplasmic fractions revealed that the unconjugated form of ISG15 localized only in the cytoplasmic fraction of cells treated with Nefw.t (Figure 6A,B). Instead, unconjugated ISG15 was also detectable in the nuclear fraction in cells treated with type I or -III IFN, although in a lesser amount with respect to the cytoplasmic fraction. As expected, type II IFN (i.e., IFN-γ) did not induce ISG15.

Since STATs are typically activated in a few minutes because of the JAK phosphorylation that follows the engagement of different cytokines, chemokines or growth factors to their specific receptors, overall, these results suggest that GEN2.2 cells are stimulated by wild type Nef to release/produce activating cytokines, as do macrophages treated with Nef [18,19]. Since myrNef_SF2_4EA is not able to induce P-STAT1 and P-STAT2, and not even IRF-1 or ISG15 production, the acidic domain must have a crucial role in the effect induced by the viral protein, as we have previously observed also in human macrophages [18,19].

### 3.5. Wild-Type Nef Induces the Production of a Different Pattern of Cytokines/Chemokines in GEN2.2 Cells Compared to Differentiated THP-1 Cells

To identify the largest number of cytokines/chemokines secreted in response to Nef treatment, the Bio-Plex Pro Human Cytokine 27-Plex Immunoassay, able to detect up to 27 cytokines, was used. To this aim, GEN2.2 cells were or were not treated with myrNef_SF2_w.t and myrNef_SF2_4EA, and a time course analysis was also performed on supernatants harvested within a 20 h interval to evidence possible time differences in the release of the analysed cytokines/chemokines (Figure 7A).

Early on (i.e., 4–6 h), wild type Nef induced the production of chemotactic factors e/o pro-inflammatory mediators, such as MCP-1 (1.54-fold vs. Ctrl), TNF-α (2.07-fold vs. Ctrl) and IL-8 (30.40-fold vs. Ctrl) and the growth factor G-CSF (4.23-fold vs. Ctrl). Later, i.e., after 20 h, a significant increase was also observed for IP-10 (9.05-fold vs. Ctrl) and MIP-1β (5.13-fold vs. Ctrl). On the other hand, the Nef 4EA mutant did not significantly increase the production of any of the analysed cytokines/chemokines (Table 1).

We also evaluated whether and to what extent the cytokines/chemokines released in response to Nef treatment by GEN2.2 cells differed with respect to those secreted by THP-1 cells used as a model of human macrophages, a cell type widely known as one of the major reservoirs of HIV (Figure 7B). Our research group already observed that, in primary macrophages, Nef induced pro-inflammatory cytokines such as MIP-1β, IL-6, IL-1β, TNF-α and IFN-β [21]. However, in this context, we analysed the expression of a greater panel of cytokines/chemokines on the THP-1 monocytic cell line differentiated with PMA. It is noteworthy that the profile induced by Nef in GEN2.2 cells was different from that observed in differentiated THP-1 (Table 2). The viral protein stimulated THP-1/PMA to release IP-10, IL-8 and MIP-1β, although at a different extent compared to GEN2.2 cells. Unlike GEN2.2 cells, TNF-α was strongly induced by Nef treatment in THP-1/PMA cells (200-fold vs. Ctrl). In addition, Nef also increased IFN-γ, RANTES, IL-15, FGF basic and MIP-1α. Interestingly, not only did the two cell lines present a different panel of cytokines/chemokines induced by Nef, but they differed also in the amount produced by control cells. In particular, the basal level of TNF-α was high in GEN2.2 cells with respect to THP-1 cells, and the opposite was true for G-CSF, IL-8 and IL-1ra. Regarding the production of type I or III IFN, the only known cytokines able to induce the tyrosine phosphorylation of STAT2, we were unable to detect their production due to the low sensitivity of the cytokine array used (see discussion).

### 3.6. Supernatant of GEN2.2 Cells Treated with myrNef_SF2_w.t Stimulates an Early Response of Untreated GEN2.2 Cell Population Independently from the Extracellular Vesicle (EV) Content

During HIV infection, pDCs are exposed to the local microenvironment influenced by infected cells. In this context, their activation is not necessarily caused by the virus, but it could be the consequence of the interaction with immunostimulatory molecules in the intercellular space. Hence, to determine whether the cytokine/chemokine milieu in the supernatants of Nef-activated GEN2.2 cells was sufficient to stimulate fresh cells not previously treated with the viral protein, new untreated GEN2.2 cell populations were cultured with supernatants collected from GEN2.2 cells, stimulated or not stimulated with myrNef_SF2_w.t for 20 h. As shown in Figure 8A,B, medium conditioned by Nef-treated GEN2.2 cells resulted in earlier activation of STAT1, after only 30 min. The fact that the activation occurred more rapidly than by following Nef treatment (3 h, as reported above) excluded the possibility that residual Nef in the supernatants was responsible for this effect. These data enforce the results previously reported, confirming the capacity of Nef to act on pDCs by promoting the release of cytokines/chemokines involved in STAT1 activation and shows that pDCs are also promptly responsive to the surrounding extracellular milieu.

Although cytokines are generally thought to exert biologic influence as soluble molecules, several cytokines have been reported to be associated with EVs, such as a membrane bound form of TNF-α, chemokines associated with lipid rafts, or cytokines, such as the IL-1 family, which lacks a signal peptide for secretion through the classical pathway [50,51]. Moreover, EV-associated cytokines became biologically active upon interacting with sensitive cells, thus representing an important system of cell–cell communication in both health and disease. In this regard, in HIV-infection it was shown that the amount of EV-associated cytokines was increased [52]. Considering these recent observations, we wondered whether GEN2.2 cells would have responded in the same way after treatment with supernatants collected from treated cells but depleted of EVs. To this aim, EVs were cleared or not from supernatants collected from GEN2.2 cultures treated or untreated for 20 h with myrNef_SF2_wt by differential ultracentrifugation, and then used to treat new GEN2.2 cultures (Figure 8C,D). The depletion of EV content did not significantly affect the cell response. Indeed, supernatants depleted of EVs maintained the capacity to activate STAT1 tyrosine phosphorylation early, after only 30 min, thus suggesting that most STAT1 activating factors must be secreted in free form and be primarily responsible for the early activation observed.

### 3.7. Set Up of the Protocol for GEN2.2 Cell Labelling with Bodipy C16

Emerging evidence has reported on the important role of EVs in the intercellular communication in both physiological and pathological conditions, including HIV infection [53,54,55]. Thus, the production of EVs was investigated. Considering the relevant number of cells necessary to isolate a good quantity of EVs and the already known difficulty in isolating sufficient amounts of primary pDCs, we set up the protocol again using the GEN2.2 pDC-like cell line. To characterize and quantify the vesicles produced by GEN2.2 cells and study how their release could be modulated in response to Nef stimulus, we adopted a methodology, developed by Sargiacomo and colleagues [41], based on cell treatment with the commercially available Bodipy C_16_ fatty acid. This latter, upon uptake by the cells, entered the cellular lipid metabolic pathway without affecting the natural lipid metabolism or perturbing the lipid homeostasis inside the cell [41]. As a result, labelled cells released small and medium/large vesicles (hereafter respectively referred to as exosomes and microvesicles) that, being fluorescent, could be examined and quantified with conventional flow cytometry.

To define the optimal conditions for GEN2.2 treatment with the fluorescent lipid, pulse-chase experiments were performed. Firstly, cells were pulsed with different concentrations of Bodipy C_16_ for different times and analysed with confocal microscopy and flow cytometry. As shown in Figure 9A, the fluorescent probe was taken up by cells very rapidly, just after 15 min, and its uptake increased during pulse times. Remarkably, Bodipy C_16_ became more and more concentrated over time in the perinuclear area corresponding to the endoplasmic reticulum (ER). Regardless of concentration, we observed that Bodipy C_16_ uptake reached a plateau between 1 and 3 h, thus, a time of 2 h was chosen for cell labelling (Figure 9B). However, we did not identify a concentration limit, because, regardless of the time treatment used, cells showed a linear uptake, suggesting a capability to further internalize the fluorescent lipid even at higher concentrations. Therefore, for the subsequent analyses, we decided to select the two highest concentrations (2.5 and 3.5 µM) whose mean fluorescence intensity (MFI) reached high values.

Since our interest was in collecting EVs after 20 h, we also verified how long the fluorescence persisted inside the cells after Bodipy treatment. To this end, GEN2.2 cells were pulsed with 2.5 and 3.5 µM of Bodipy for 2 h; afterwards, cells were washed to eliminate the residual fluorescent probe and fresh medium supplemented with 10% FBS was added. GEN2.2 cells were then chased for different times up to 24 h and observed by confocal microscopy. Cell fluorescence appeared more and more diffuse with few spots that were mostly chased out after 24 h (Figure 10A).

During the chase times in fresh medium, GEN2.2 cells treated with 3.5 µM of Bodipy C_16_ showed a drastic reduction in cell fluorescence, by about 80% after only 1 h, whereas it slowly decreased afterwards (Figure 10B). However, the fluorescence was still detectable up to 24 h, ensuring that vesicles would have been able to incorporate the fluorescent lipid throughout the period of their production. A similar pattern was observed when treating cells with 2.5 µM; although after 1 h, cells presented a reduction in the fluorescence of 68% compared to initial values, their fluorescence intensity was slightly lower than in cells treated with 3.5 µM. Therefore, the concentration of 3.5 µM was chosen for cell labelling.

In conclusion, the reported data indicate that Bodipy C_16_ is internalized by GEN2.2 cells, reaches a plateau at 2 h and, although the fluorescence undergoes a rapid reduction, it does not chase out completely after 24 h. The reduction in fluorescence observed is consistent with the idea that the fluorescent lipid, once transported to the ER where it is mainly metabolized in phospholipids, is then directed to the endosomal pathway and released into the extracellular milieu as part of the EV membrane.

### 3.8. Nef Reduces the Exosome Production and Is Found Associated with the Exosomal Fraction

The above-described methodology based on cell treatment with Bodipy C_16_ allowed the detection and count of EVs through conventional FC, overcoming the problem of the reduced size of exosomes (below 200 nm). Indeed, although it is well known that the detection of vesicles or particles smaller than 300 nm by FC based on light-scattering is severely hampered by noise events, the novel strategy allowed the discrimination of fluorescently labelled vesicles from non-fluorescent noise by coupling the fluorescent signal of vesicles with the light-scattering.

Therefore, GEN2.2 cells were pre-treated with 3.5 µM Bodipy for 2 h in complete medium supplemented with 0.3% FBS. Afterwards, cells were washed to remove residual lipid, and fresh medium supplemented with 10% ultracentrifugated FBS and containing myrNef_SF2_w.t was added (Figure 11A). Fluorescent exosomes and microvesicles released in medium were isolated after 20 h by differential ultracentrifugations and then processed for FC analysis (Figure 11B,D). Interestingly, in comparing Bodipy-stained exosomes’ secretion in cells treated with the viral protein with respect to the untreated ones, the production of exosomes turned out to be reduced by about 40% in response to Nef stimulus, whereas that of microvesicles did not appear to be influenced (Figure 11C,E).

According to the guideline published in 2018 by the Journal of Extracellular Vesicles [42], to better characterize the nature of the isolated vesicles, we analysed at least one of the transmembrane proteins (CD81) and cytosolic proteins (TSG101, ALIX, HSP90 and Flotillin-1) commonly found in mammalian cell-derived EVs. Furthermore, we evaluated the presence of COXIV, a protein localized in mitochondria, which *a priori* is not enriched in the smaller EVs (<200 nm diameter) of the plasma membrane or endosomal origin. All specific exosomal markers were present in the sample corresponding to exosomes but not in the microvesicular one, whereas, as expected, COXIV was detected only in cellular lysates (Figure 11F). This analysis formally confirmed the nature and the purity of the isolated vesicles and allowed, for the first time, one peculiar aspect of exosomes isolated from these pDC-like cells to be revealed, i.e., the low expression level of the tetraspanin CD81, whose detection required longer exposure time. The lower expression in the exosomal samples mirrors the low intracellular expression of CD81, which has also been recently reported in human primary pDCs [56], and that distinguishes pDCs from most of other cell types, including myeloid DCs.

Based on what was reported in the literature regarding the ability of Nef to be transferred to uninfected cells through EVs, we wondered whether the recombinant viral protein followed the same destiny of the viral protein when endogenously expressed in HIV-infected cells [27]. As shown in Figure 11F, Nef protein was found to be associated with EVs, similarly to the protein endogenously expressed during HIV infection. Of note, we also observed that Nef is preferentially found in the exosomal fraction, whereas it was undetectable in the microvesicular one. The specificity of the observed signal was confirmed by the absence of the band corresponding to Nef protein in exosomes isolated from untreated cells. Moreover, as expected, Nef was also detected in the cellular extract, confirming its internalization into GEN2.2 cells during the treatment.

## 4. Discussion

In the last years, remarkable progress has been made regarding pDCs’ functions in HIV infection, especially thanks to studies conducted on HIV-infected pDCs. The results presented here highlight a new potential effect that the extracellular HIV-1 Nef protein could play on this dendritic cell subset. The virulence properties of HIV-1 Nef have, for a long time, been mainly associated with its multiple biochemical activities inside infected cells acting as a molecular adaptor, but in the new millennium, the attention has been focused on the effects that the viral protein exerts on bystander uninfected cells, where it can be transferred through different mechanisms, including cell-to-cell contacts, nanotubes and EVs [23,24,25,33,57]. In addition, in vivo soluble Nef could be released by necrosis of infected cells and its presence in free form (i.e., non EV-associated) in the plasma is suggested by the production of specific antibodies against this viral protein, present in infected individuals. In the case of pDCs, the contact with extracellular Nef or Nef transfer from infected cells could occur in proximity with infected mucosal sites, where they migrate in response to inflamed conditions, or in lymph nodes.

In the present work, we investigated cytokine production, intracellular signalling and the EV production induced by myrNef_SF2_ treatment of the human plasmacytoid dendritic GEN2.2 cell line. This cell line was used as a model of pDCs to overcome the problems related to the low amount of primary pDCs in blood (0.2–0.5% of PBMC) and the large culture volumes necessary to isolate enough EVs for biochemical analyses.

The rationale for carrying out this investigation was laid out by preliminary, promising results, indicating that: (i) 4 h of cellular treatment with myrNef was able to induce STAT1 tyrosine phosphorylation in PBLs purified by PBMCs isolated from human healthy donors, but not in PBLs depleted of pDCs (Figure 1); (ii) about 30% of primary pDCs internalized the recombinant Nef protein (Figure 2) and (iii) the exogenous treatment of primary pDCs up-regulated the expression of *mxA* and the IRF-7 transcription factor, two proteins codified by IFN-inducible genes, whose up-regulation is usually used as a surrogate marker for IFNs’ production (Figure 1 and Figure 3). In addition, a partial nuclear translocation of the transcription factor IRF-7 was also observed.

Regarding the response of GEN2.2 cells, here we report that myrNef is internalized by GEN2.2 cells, but less efficiently than we previously observed in primary monocyte-derived macrophages (MDMs) [19,58], whereas the myrNef_SF2_ protein, as well as its mutant 4EA, was rapidly and efficiently internalized in most MDMs (see Figure 2C in [19]). The different efficiency might be attributed to the lower phagocytic/internalization ability that distinguishes this particular cell line from macrophages. In respect to the entry mechanism, experiments were performed in GEN2.2 cells using different inhibitors of the entry process (data not shown), but the results were not conclusive because none of the tested inhibitors was able to prevent Nef internalization. Moreover, Nef induces in GEN2.2 cells the tyrosine phosphorylation of both STAT1 and STAT2 proteins starting from 3 h of treatment and substantially influences the gene expression program regulated by STAT1 and 2 activation, as indicated by the later induction of IRF-1, STAT1 and ISG15, codified by three IFN regulated genes. Conversely, the Nef mutant 4EA, while it is internalized similarly to the wild type protein, is unable to induce the same effects, highlighting the importance of the N-terminal acidic domain E^66^EEE^69^ in the signalling pathway induced by the protein. These results confirm and add relevance to our previous findings obtained in primary macrophages [18,19]. We can infer that GEN2.2 cells are less sensitive to Nef treatment with respect to primary macrophages. Indeed, in in vitro culture of MDMs, myrNef induced the STAT1/2 phosphorylation in response to the release of a set of cytokines and chemokines (CCL2/MIP- 1α, CCL4/MIP-1β, IL-6, TNF-α, IL-1β and IFNβ) with lower concentrations of the viral protein (i.e., 10–100 ng/mL) [18,20,21] compared to GEN2.2 cells (i.e., 300 ng/mL). Moreover, the activation of STAT1 and 2 was observed earlier (after only 2 h of cell treatment) than in GEN2.2 cells, where it starts from 3 h of Nef treatment.

STAT activation is the consequence of the production of activating factors, also including some IFN types, as suggested by the induction of STAT2 tyrosine phosphorylation induced only by type I or type III IFN signal transduction pathway activation. Using a Bio-Plex Pro Human Cytokine 27-Plex Immunoassay able to detect up to 27 cytokines, we observed in GEN2.2 cells that myrNefwt induced, at an early time of cell treatment (i.e., after 4 h), the production of chemotactic factors e/o pro-inflammatory mediators, such as MCP-1 (1.54-fold vs. Ctrl), TNF-α (2.07-fold vs. Ctrl) and IL-8 (30.40-fold vs. Ctrl), and the growth factor G-CSF (4.23-fold vs. Ctrl). Later (i.e., after 20 h of cell treatment) a significant increase was also observed for IP-10 (9.05-fold vs. Ctrl) and MIP-1β (5.13-fold vs. Ctrl). On the other hand, the Nef mutant 4EA did not significantly increased the production of any of the analysed cytokines/chemokines. These results are consistent with available literature concerning primary pDCs, which, in addition to IFNs, are known to produce a number of inflammatory cytokines and chemokines, including TNF-α, MIP-1α, MIP-1β, RANTES, IL-8 and IP-10 [6].

In addition to the Bio-Plex Pro Human Cytokine 27-Plex Immunoassay (Bio-Rad), we also used the VeriPlex^TM^ human IFN 9-plex ELISA Kit (PBL Assay Science, Piscataway, NJ, USA) detecting nine cytokines (IFN-α, IFN-β, IFN-γ, IFN-λ, IFN-ω, IL-1α, IL-6, IP-10 and TNF-α) to measure the production of different IFN types. Unfortunately, the sensitivity of the array for the different IFN types, based on the standard curves, was too low and did not allow their detection, because it required a production of at least 400 pg/mL (about 400 IU/mL) for IFN-α, 2400 pg/mL for IFN-β (about 2400 IU/mL), 630 pg/mL (about 63 IU/mL) for IFN-γ and 1500 pg/mL for IFN-λ. Therefore, we concluded that the amount of IFN production able to activate STAT2 (i.e., type I, type III or both) was below the detection limit of the kit.

Unlike GEN2.2 cells, Nef also induces the release of IFN-γ, RANTES, IL-15, FGF basic and MIP-1α in THP-1/PMA cells. The induction of MIP-1β, MIP-1α and TNF-α production in THP-1/PMA cells is in agreement with the previous observations reported in primary macrophages by our research group [21]. Together, the data (see Figure 7 and Table 1 and Table 2) highlight the ability of extracellular Nef to induce the release of a different pattern of cytokines/chemokines according to the cell type, probably contributing to fuel, in different ways, the intense “cytokine storm” that characterizes HIV infection [59]. For instance, the monocyte chemoattractant protein-1 (MCP-1/CCL2), by regulating migration and infiltration of monocytes/macrophages [60], could strongly favors monocyte recruitment in HIV infection sites. In the same manner, the release of the chemokines MIP-1α and -1β (i.e., CCL3 and CCL4) and IL-8 could favor the recruitment and activation of CD4^+^ T cells. In accordance with these results, Li et al. reported that, in the SIV-macaque model, the vaginal exposure to SIV induces the local mucosal pDC activation, resulting in the early production of IFN-α and chemokines (MIP-1α and -1β), which is followed by CCR5^+^CD4 T cell recruitment, a mechanism by which pDCs could fuel SIV replication [7]. However, Carmona-Saez showed that pathways related with transcription regulation, signal transduction and intracellular traffic were consistently down-regulated in GEN2.2 cells compared with pDCs [46]. Therefore, it is necessary to consider that the pattern of cytokines/chemokines and gene expression levels could be somewhat different in this cell line compared to primary pDCs.

Regarding the chemokine IP-10/CXCL10 (also known as interferon-γ induced protein 10 kDa), it is mainly produced in response to IFN γ [61], however, the induction of IP-10 has been also reported in response to treatments with IFN-β [62] or IFN-α2a [63]. IP-10 plasma levels have been demonstrated to be abnormally increased after HIV infection and tightly associated with HIV disease progression [64]. IP-10 high levels in human cervical and colonic mucosal tissue have been described to correlate with the recruitment of HIV target cells to the mucosa surface, thus facilitating the transmission process [65]. They may also impair immune cell functions (T cells and NK cells) and promote HIV replication and latency [64]. Regarding the possible mechanisms underlying the increase in IP-10, the involvement of a combination of HIV particles or HIV proteins, such as Tat and TLR7/9 [64,66,67], has been hypothesized. Here, we described the ability of Nef protein alone to induce IP-10 expression in our in vitro model of uninfected macrophages (THP-1 cell line) and pDCs (GEN2.2 cell line). Since Nef stimulates the release of TNF-α in GEN2.2 cells, we can hypothesize that the mechanisms underpinning IP-10 production induced by Nef could involve the cooperation among this cytokine and the activation of JAK/STAT1 and the NFκB signalling pathways. Even the late production of ISG15 could contribute to IP-10 expression, since it has been reported that elevated levels of this IFN-induced protein can effectively promote IP-10 expression in macrophages, because ISG15 decreases the inhibitory effects exerted by microRNA-21 on IP-10 production [68].

The Nef-induced modification of the pattern of released cytokines/chemokines may lead to consequences on neighboring cells. To verify this, we treated fresh GEN2.2 cells with medium from GEN2.2 cells stimulated with Nef. This resulted in earlier tyrosine phosphorylation (after 30 min) of STAT1, showing that Nef-induced secretome is also able to activate this transcription factor in new pDCs, and the latter are promptly responsive to this surrounding extracellular milieu.

Emerging studies have also identified the release of EVs as a potential mechanism by which cytokines/chemokines can be secreted into the extracellular space [50,51]. To determine the influence of EV-associated cytokines, we treated GEN2.2 cells with supernatants collected from Nef-treated GEN2.2 cells and depleted of EVs. This resulted again in the early activation of STAT1, indicating that its activation is mainly due to the secretion of free activating factors. Through the release of a specific cytokines/chemokines pool, extracellular Nef could potentially make pDCs able to indirectly amplify and activate the locally available target cells for viral infection and/or influence the immune response to the infection.

Another interesting finding of our study relies on the characterization of the EV production induced by Nef protein in our pDCs model. Despite the recent expansion of studies conducted on vesicles, nowadays, there are few methods for the reliable quantification and characterization of EVs. In this study, we adopted the methodology developed by Sargiacomo and colleagues based on cell treatment with the Bodipy C_16_ fatty acid that allows the release of fluorescent EVs, thus overcoming the problem correlated to the reduced size of exosomes and their detection by means of FC instrument [41]. However, the presence of vesicles that might escape the Bodipy labelling cannot be formally ruled out because, not being fluorescent, they cannot be detected through FC, and therefore EVs released by the cells could be underestimated. Interestingly, unlike what was reported in the literature regarding other cell types endogenously expressing the viral protein, such as astrocytes or lymphocytes [25,32], Nef treatment does not increase the production of exosomes in GEN2.2 cells; conversely, a 40% reduction was observed. It is known that Nef inside the cells exploits the vesicular transport machinery of the host cell to favor its diffusion and HIV infection. In particular, Nef intracellular expression increases the number of MVBs in some cell types that could also favour the egress of viral particles in infected cells [69,70]. Regarding our pDCs model, the exogenously added protein activating the production of IFN-induced proteins, such as ISG15, could interfere with the release of vesicles. Indeed, the ISGylation of TSG101, a transmembrane protein belonging to the ESCRT complex involved in the exosome biogenesis, has been reported to interfere with exosome formation [48]. Therefore, the ISG15 expression in myrNef-treated GEN2.2 cells could negatively affect the ability of Nef to increase the release of exosomes.

Despite the consistently reported association of Nef with EVs, it still remains unclear which type of EVs are involved, since, according to the cell type, Nef was found to be associated with small or large vesicles [23,33,35]. To date, several groups have explored the cellular mechanisms associated with EV-mediated Nef secretion. The importance of a motif comprising residues 66-70 (VGFPV) in the N-terminal region of the protein, termed the secretion modification region (SMR), has been described. This region has been demonstrated to be involved in the binding of Nef to the host protein mortalin [71], resulting in its release into EVs [72]. Nevertheless, mortalin is a member of the heat shock 70-kDa protein family that associates with lipid rafts in the plasma membrane and regulates the intracellular trafficking of cell surface receptors, but, since it is present in both microvesicles and exosomes, its binding to Nef cannot be a determinant factor for its release into exosomes rather than into microvesicles. Considering the above, the specific internalization of Nef into exosomes might also require other interactions that could direct the viral protein into the endosomal pathway involved in the biogenesis of exosomes. One possible mechanism could be the direct association of this myristoylated protein with lipid rafts, which are enriched in MVBs and may lead to piggybacking of the tethered Nef protein into exosomes [73].

In vitro studies already showed that exosomes produced by infected cells play a key function in the activation of the immune response mediated by pDCs and are involved in the type I IFN production [74]. Concerning GEN2.2 cells, after cell treatment with the viral protein we found Nef both intracellularly and associated with the released exosomes collected by ultracentrifugation, but not with microvesicles.

Considering that the predominant form of Nef in circulation is probably associated with exosomes, it is important to underline that the effects described in this study might differ from those induced by Nef-containing EVs released by the intracellular expression of the viral protein. In this regard, exosomes containing HIV-1 Nef protein turned out to have multiple pathogenic effects, such as the induction of T-cell apoptosis [24] and the down-modulation of cell surface molecules (i.e., MHC-I and CD4) for immune evasion [75]. Moreover, the cellular expression of HIV-1 Nef induces the release of exosomes incorporating active ADAM17/TACE [76], a metalloprotease that promotes the maturation of pro-TNFα into its active form. Indeed, the production of TNF-α was observed in resting CD4^+^ T lymphocytes challenged with ADAM17/Nef EVs, rendering them competent for HIV-1 expression and replication [27,28,30]. In agreement with our data, it has been reported that Nef-containing EVs modulate the secretome in microglia, increasing Toll-like receptor-induced cytokine and chemokine levels (including IL-12, IL-8, IL-6, RANTES, and IL-17A) [31], and reorganize lipid rafts, potentiating an inflammatory response in bystander cells [77].

We hypothesize that in vivo soluble Nef released by necrosis of infected cells might be internalized directly or via FcRs after forming immune complexes with anti-Nef antibodies produced in infected individuals. Then, the acidic environment of the endosomal compartments could favour the release of the viral protein in free form, which might interact with the endosome membrane and, eventually, be translocated via a flip-flop mechanism by inducing signal transduction pathways involved in the regulation of the cellular secretome. This speculation might also explain the results we previously obtained in THP-1 differentiated cells, using the silencing procedure and co-immunoprecipitation techniques, indicating that (i) Nef is able to form a complex with TRAF2 through its conserved 4E acidic domain and (ii) extracellular Nef-induced production of inflammatory cytokines and IFN-β in THP-1 requires the specific intracellular adaptors TRAF2 and 6 and the 4E acidic domain [19].

Finally, Nef, free in the extracellular space, might represent a danger signal, inducing cellular response different from that of Nef transferred via nanotubes, EVs or cell-to-cell contact. This should be taken into consideration in the development of an HIV vaccine, based also on the expression/presence of this viral protein or its conserved domain. In conclusion, the results presented here lay the foundation for extending the study to primary pDCs, to identify the destiny of the internalized protein and analyse the content and the biological activity of the exosomes released by treated cells.

## Figures and Tables

**Figure 1 viruses-14-00074-f001:**
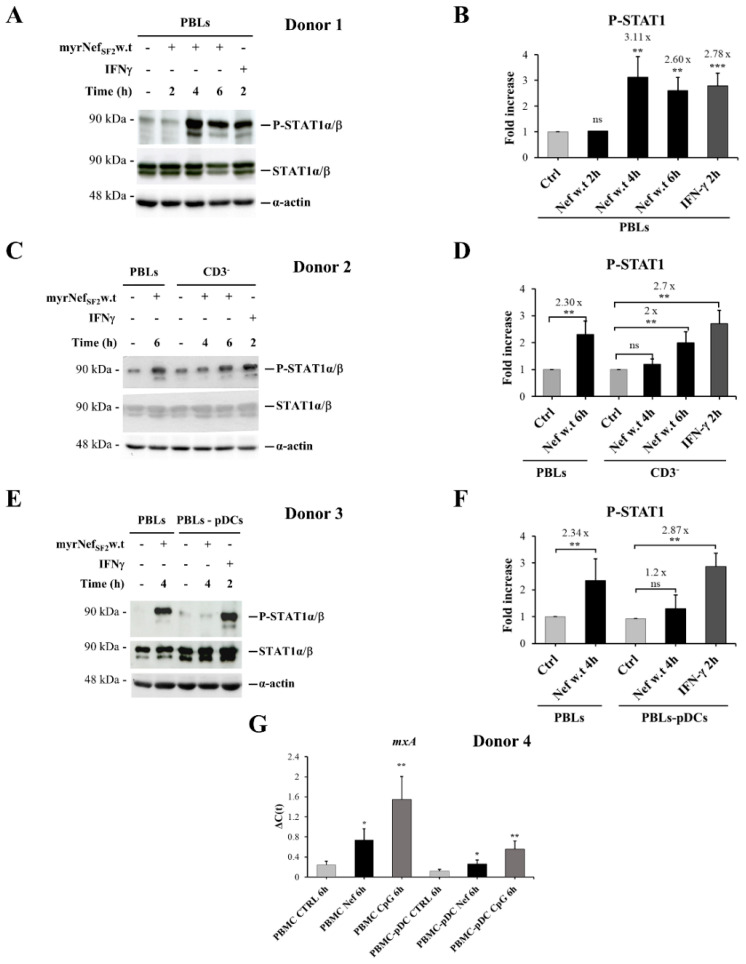
myrNef_SF2_w.t induces the tyrosine phosphorylation of STAT1 in PBLs, but not in PBLs depleted of pDCs, and increases *mxA* expression. PBLs (**A**), PBLs depleted of CD3^+^ cells (**C**) and PBLs depleted of pDCs (PBLs-pDCs) (**E**) were seeded at 4 × 10^6^ cells in a 12-well plate and treated with 300 ng/mL of myrNef_SF2_w.t for the indicated time points. The treatment with IFN-γ (15 IU/mL) was used as a positive control. Cell lysates (50 µg of proteins) were analysed in 9% SDS-PAGE gel and the immunoblotting was performed using a phospho-Tyr(701)-STAT1 specific antibody. Anti-α-actin was used as an internal control of the loaded samples. (**B**,**D**,**F**) P-STAT1 was normalized to actin by densitometric analysis and reported as fold increase compared to control. (**G**) PBMCs and PBMCs depleted of pDCs (PBMCs-pDCs) were seeded at 2 × 10^6^/2 mL and treated for 6 h with 300 ng/mL of myrNef_SF2_w.t or 1 µM of CpG A as a positive control. Ctrl: untreated cells. After treatment, cells were harvested and processed for RNA extraction. *mxA* expression was evaluated by qRT-PCR and the data were normalized using the 2-∆Ct formula, where ∆Ct represents the difference between the amplification cycles of *mxA* gene and the amplification cycles of the housekeeping gene GAPDH (glyceraldehyde-3-phosphate-dehydrogenase), constitutively expressed in all cell types. The experiments were performed using four different donors. Histograms: mean ± S.D. One-way ANOVA test; *, *p* < 0.05; **, *p* < 0.01; ***, *p* < 0.005; ns, not significant vs. respective Ctrl (untreated cells).

**Figure 2 viruses-14-00074-f002:**
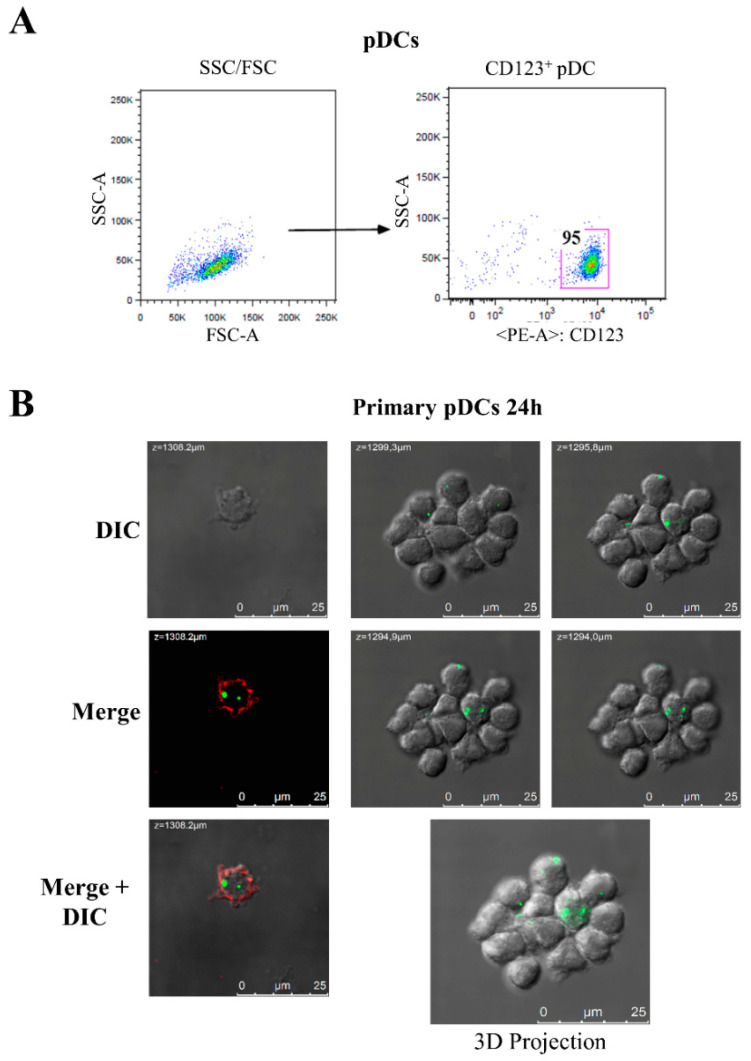
Internalization of Nef protein in plasmacytoid dendritic cells. (**A**) Flow cytometry plots showing the forward light scatter/SSC profile of the cells. Purity of pDCs was determined by staining cells with anti-CD123 antibodies. (**B**) Primary pDCs were seeded at 10^5^ cells/200 µL in 96-well plates. Purified pDCs were treated with 300 ng/mL of myrNef_SF2_ conjugated with AlexaFluor488 (green) for 24 h. Afterwards, cells were fixed, as reported in Materials and Methods, and analysed by confocal microscope (Leica TCS SP5), software LAS AF version 1.6.3 (Leica Microsystems). Plasma membrane counterstaining was performed using PKH26-GL (red). Objective 63.0X. DIC: Differential Interference Contrast Images. Scale bars 0–25 µm. Representative images of two independent experiments are shown.

**Figure 3 viruses-14-00074-f003:**
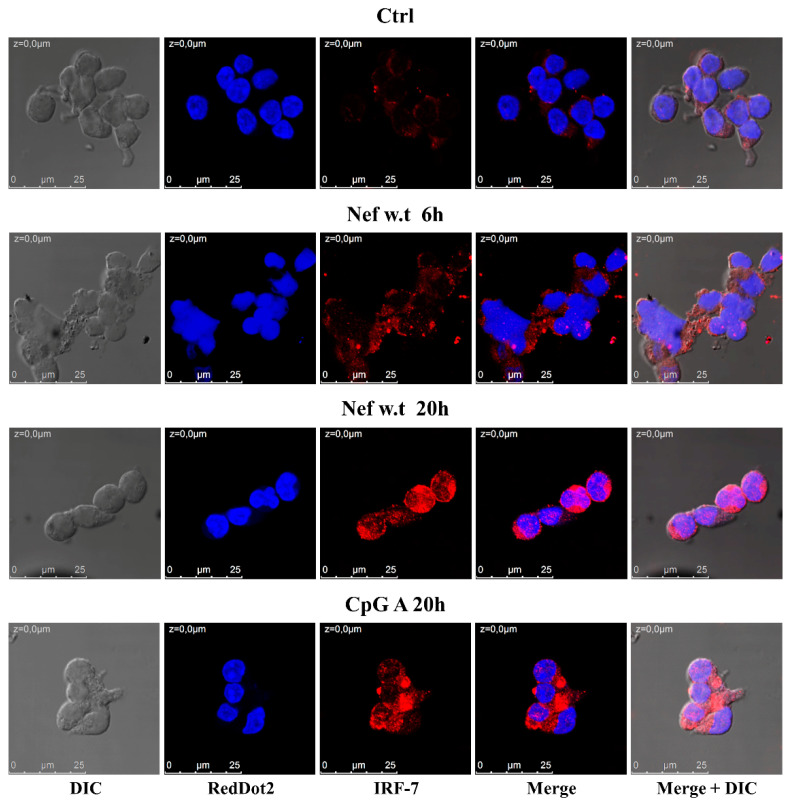
IRF-7 is upregulated and translocates to the nucleus after treatment with Nef protein. 0.5 × 10^5^ pDCs were treated for 6 h and 20 h with 300 ng/mL of myrNef_SF2_w.t or for 20 h with CpG A (1 µM), as a positive control. Ctrl: untreated cells. Cells were afterwards fixed in PFA 4%, permeabilized and incubated with anti-IRF-7 antibody and with a secondary antibody conjugated with AlexaFluor546 (red), as reported in Materials and Methods. Nuclei (blue) were stained using the dye RedDot2. Images were acquired with the confocal microscope Leica TCS SP5 and processed using the software LAS AF version 1.6.3 (Leica Microsystems). Objective 63.0X. DIC: Differential Interference Contrast. Scale bars 0–25 µm. For further details see Materials and Methods section.

**Figure 4 viruses-14-00074-f004:**
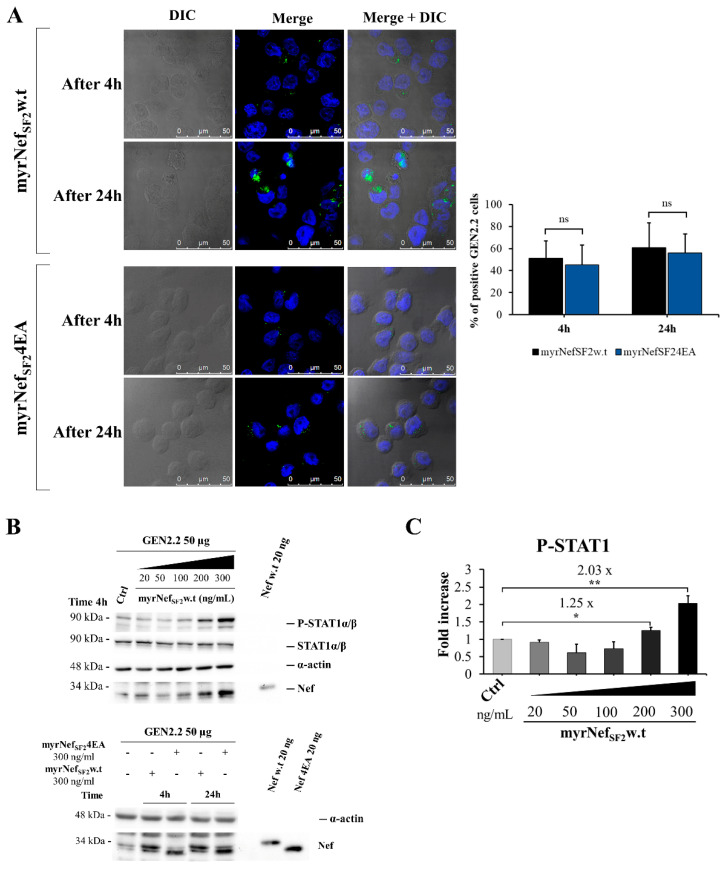
Internalization of Nef protein by GEN2.2 cells. (**A**) Confocal microscopy analysis of GEN2.2 cells seeded at 0.1 × 10^6^ cells/150 μL and treated for 4 h and 24 h with 300 ng/mL of myrNef_SF2_w.t and myrNef_SF2_4EA conjugated with AlexaFluor488 (green). Afterwards, cells were placed on a microscope slide and fixed in PFA 4%. Samples were mounted with Vectashield antifade mounting medium containing DAPI to visualize nuclei (blue). Images were acquired with the confocal microscope Leica TCS SP5 and processed with the software LAS AF version 1.6.3 (Leica Microsystems). Objective 63.0X. DIC: Differential Interference Contrast. Scale bars 0–50 µm. The images are representative of two independent experiments. Graph reporting the % of GEN2.2 cells that internalize myrNef_SF2_w.t or myrNef_SF2_4EA is shown on the right. (**B**) Representative examples of three Western blot analyses are shown. 2 × 10^6^ cells were treated with increasing concentrations of myrNef_SF2_w.t for 4 h (upper panel) and with 300 ng/mL of myrNef_SF2_w.t or its mutant 4EA for 4h and 24 h (lower panel). Cell lysates (50 µg) were resolved on 11% SDS-PAGE gel and the immunoblotting was performed using a phospho-Tyr (701)-STAT1 and Nef specific antibody. Anti-α-actin was used as internal control of the loaded samples. (**C**) Densitometric analysis of three independent Western blotting experiments, whose representative example is reported in panel B. The band density ratios of P-STAT1 normalized relative to actin levels are reported on the graph. P-STAT1/actin ratio in control cells was set to 1. Fold increases in P-STAT1 after the addition of Nef were calculated and reported as means ±S.D. One-way ANOVA test; *, *p* < 0.05; **, *p* < 0.01; ns, not significant vs. respective Ctrl (untreated cells).

**Figure 5 viruses-14-00074-f005:**
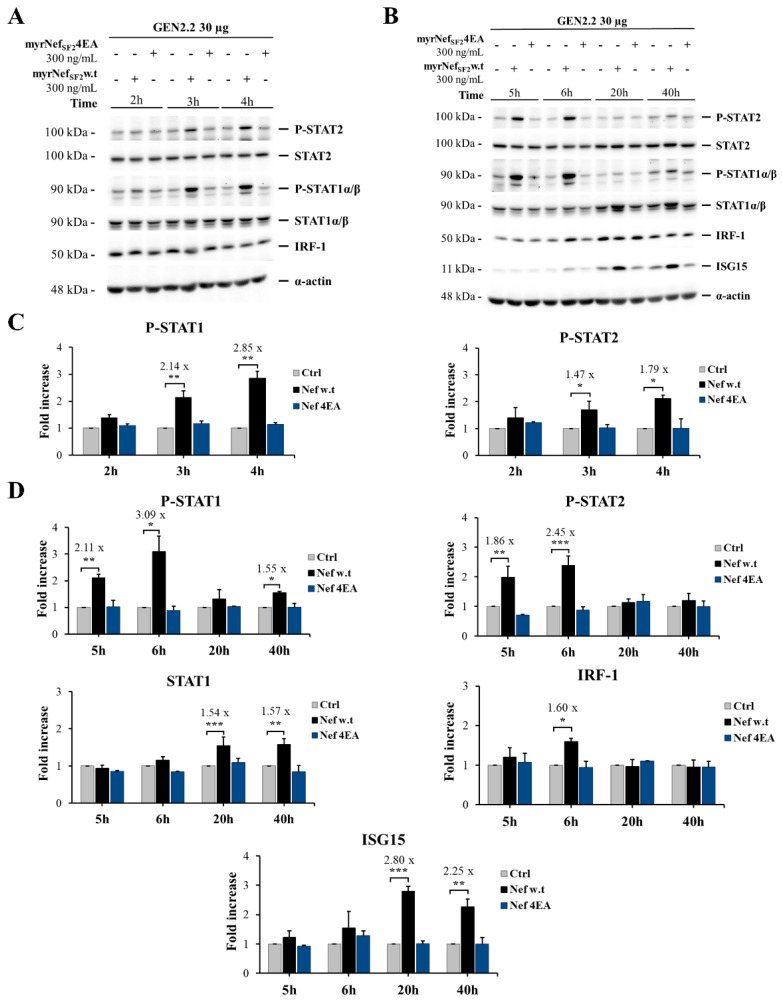
HIV-1 myrNef_SF2_w.t protein induces STAT1 and 2 activation and increases expression of proteins transcriptionally regulated by their activation (i.e., STAT-1, IRF-1 and ISG15). GEN2.2 cells were seeded at 2 × 10^6^ cells/each sample in a 24-well plate and treated with 300 ng/mL of myrNef_SF2_w.t or myrNef_SF2_4EA or left untreated (Ctrl) using two different time-courses: the shorter one (2–4 h) represented in panels A and C, and the longer one (5–40 h) reported in panels B and D. Cells were lysed and 30 µg of proteins of each cell extract were run on 9–13.5% SDS-PAGE gel. (**A**,**B**) Representative examples of three independent Western blots are shown. Anti-α-actin was used as an internal control of the loaded samples. (**C**,**D**) Densitometric analyses of three independent Western blotting experiments. The band density ratios of P-STAT1, P-STAT2, ISG15 and IRF-1, normalized relative to actin levels, are reported in the histograms. P-STAT1, P-STAT2, ISG15 and IRF-1/actin ratios in control cells (Ctrl) were set to 1. Fold increases of each analysed protein were calculated and reported as means ± S.D. One-way ANOVA test; *, *p* < 0.05; **, *p* < 0.01; ***, *p* < 0.005 vs. respective Ctrl.

**Figure 6 viruses-14-00074-f006:**
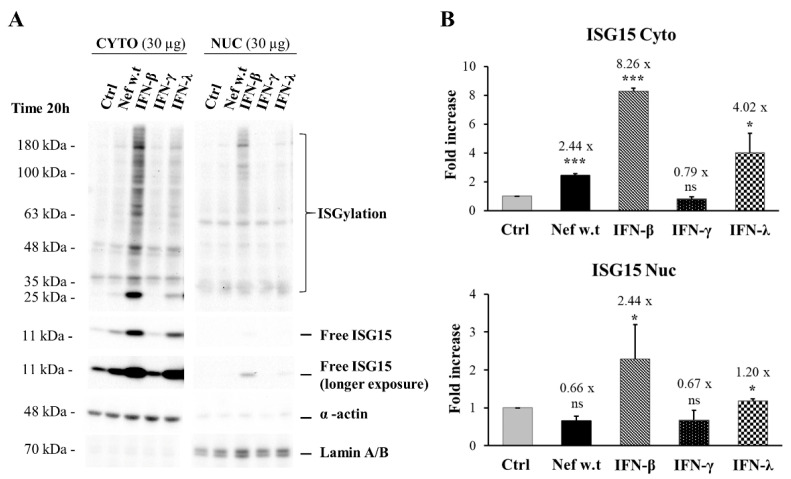
HIV-1 myrNef_SF2_w.t protein induces the production of ISG15, which is mainly localized in the cytoplasmic fraction. A total of 4 × 10^6^ GEN2.2 cells were treated with myrNef_SF2_w.t (300 ng/mL), IFN-β (1000 IU/mL), -γ (100 ng/mL) or –λ1/λ2 (100 ng/mL) or left untreated (Ctrl) for 20 h. Cells were lysed and 30 µg of proteins for each sample were run on 9–13.5% SDS-PAGE gel. (**A**) A representative Western blot is shown. (**B**) Densitometric analyses of three independent Western blotting experiments are shown. The band density ratio of free ISG15 in the cytoplasmic (Cyto) and nuclear (Nuc) fraction, normalized to relative actin or lamin A/B, respectively, are reported in the histograms. ISG15/actin ratio in control cells (Ctrl) was set to 1. Fold increases after the addition of the indicated treatments was calculated and reported as means ± S.D. One-way ANOVA test; *, *p* < 0.05; ***, *p* < 0.005, ns = not significant vs. respective Ctrl.

**Figure 7 viruses-14-00074-f007:**
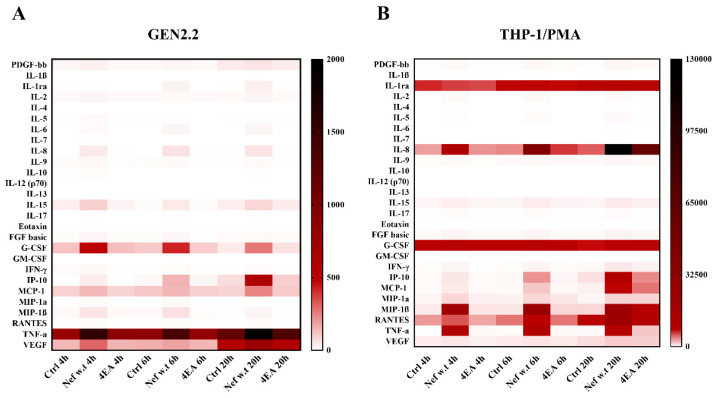
Profile of cytokines/chemokines released by GEN2.2 cells and THP-1/PMA in response to Nef treatment. GEN2.2 cells were seeded at the density of 1 × 10^6^/mL in a 24-well plate, whereas THP-1/PMA cells were seeded at 100,000 cells/cm^2^ in a 6-well plate. Both cell types were or were not treated with 300 ng/mL of myrNef_SF2_w.t or myrNef_SF2_4EA in a final volume of 2 mL. Supernatants were collected after 20 h, centrifuged at 290× *g* for 3 min to remove cells and analysed with Bio-Plex Pro Human Cytokine 27-Plex Immunoassay. (**A**) Heat map of cytokines/chemokines released by GEN2.2 cells. The color scale range is 0–2000 pg/mL. (**B**) Heat map of cytokines/chemokines released by THP-1/PMA cells. The color scale range is 0–130,000 pg/mL. Each row represents a cytokine/chemokine, whereas each column represents a sample. The values mapped are the mean of two independent experiments evaluated in triplicate.

**Figure 8 viruses-14-00074-f008:**
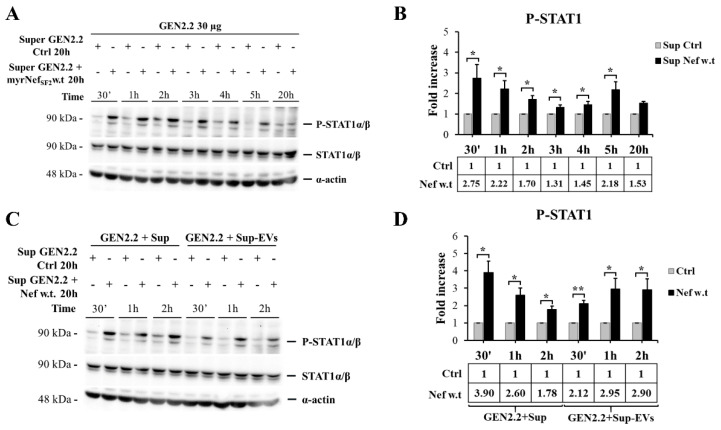
Supernatants from Nef-treated GEN2.2 cells depleted or not depleted of EVs activate STAT1 tyrosine phosphorylation of GEN2.2 cells early. GEN2.2 cells were seeded at 1 × 10^6^ cells/mL in 75 cm^2^ flask in 12 mL of final volume and left untreated or treated with 300 ng/mL of myrNef_SF2_w.t. After 20 h, supernatants from control and treated GEN2.2 cells were harvested and depleted or not depleted of EVs by ultracentrifugation. Complete supernatants (**A**) and supernatants depleted of EVs (**C**) were used to treat fresh GEN2.2 cells for the indicated time points. (**A**,**C**) Cell lysates (30 µg) were analysed on 9% SDS-PAGE gel, and the immunoblotting was performed using a phospho-Tyr (701)-STAT1 specific antibody. (**B**,**D**) Densitometric analyses of three independent Western blotting experiments are shown. The band density ratio of P-STAT1 normalized to relative actin is reported in the histograms. P-STAT1/actin ratio in control cells (Ctrl) was set to 1. Fold increases after the addition of the indicated treatments were calculated and reported as means ± S.D. One-way ANOVA test; *, *p* < 0.05; **, *p* < 0.01; vs. respective Ctrl.

**Figure 9 viruses-14-00074-f009:**
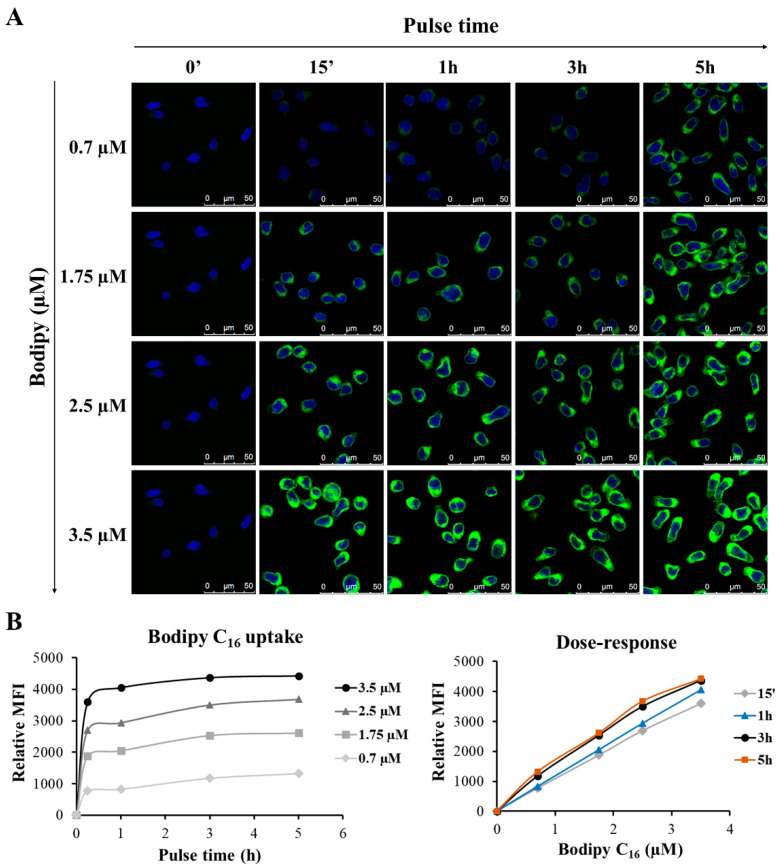
Bodipy C_16_ uptake by GEN2.2 cells. A total of 0.3 × 10^6^ GEN2.2 cells were pulsed for different times with different concentrations of Bodipy C16 (green), as indicated in the figure. (**A**) For confocal microscopy analysis, cells were placed on a microscope slide and fixed in PFA 4%. To visualize nuclei (blue), GEN2.2 cells were stained with DAPI. Images were acquired with the confocal microscope Leica TCS SP5 and processed with the software LAS AF version 1.6.3 (Leica Microsystems). Objective 63.0X. DIC: Differential Interference Contrast. Scale bars 0–50 µm. (**B**) Cell fluorescence was analysed by FC and expressed as relative MFI (mean fluorescence intensity). A representative experiment, out of three independent experiments that yielded similar results, is shown.

**Figure 10 viruses-14-00074-f010:**
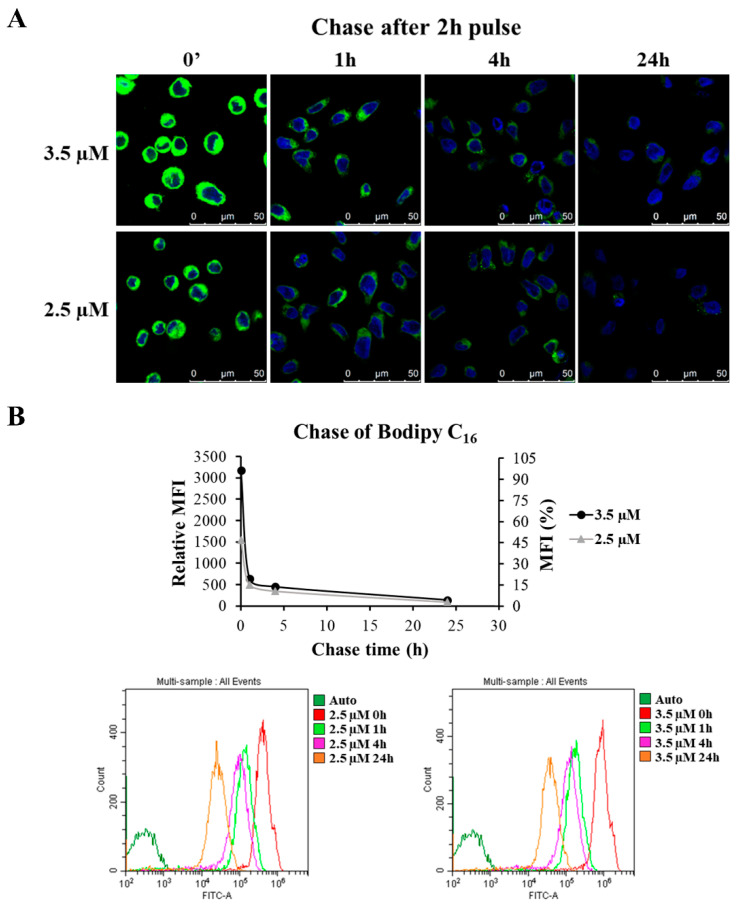
Chase of Bodipy C_16_ after 2h pulse. 0.3 × 10^6^ GEN2.2 cells were pulsed for 2 h with 2.5 µM or 3.5 µM of Bodipy C_16_ in complete medium supplemented with 0.3% FBS. Afterwards, cells were washed and chased in complete medium containing 10% FBS according to the times reported in the figure. (**A**) For confocal microscopy analysis, cells were placed on a microscope slide and fixed in PFA 4%. To visualize nuclei (blue), GEN2.2 cells were stained with DAPI. Images were acquired with the confocal microscope Leica TCS SP5 and processed with the software LAS AF version 1.6.3 (Leica Microsystems). Objective 63.0X. DIC: Differential Interference Contrast. Scale bars 0–50 µm. (**B**) Cell fluorescence was analysed by FC and reported as relative MFI and percentage of MFI (upper panel). The corresponding flow cytometry plots were reported in the lower panel. A representative experiment, out of three independent experiments that yielded similar results, is shown.

**Figure 11 viruses-14-00074-f011:**
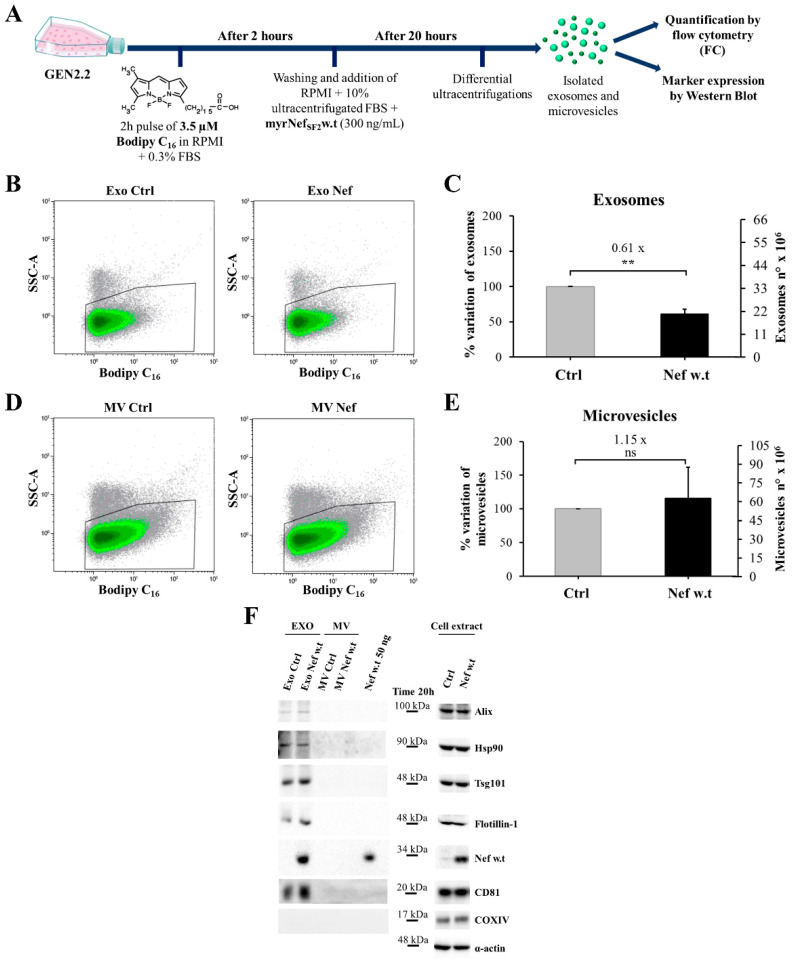
HIV-1 myrNef_SF2_ protein affects the exosome but not the microvesicle production by GEN2.2 cells and is found associated with the exosomal fraction. (**A**) Workflow of isolation of fluorescent exosomes and microvesicles from GEN2.2 cells treated with Nef protein. A total of 10^7^ GEN2.2 cells were seeded in 75 cm^2^ flask and treated for 2 h with 3.5 µM of Bodipy C_16_ in 5 mL of complete medium supplemented with 0.3% FBS. Then, cells were washed, and 12 mL of fresh medium supplemented with 10% ultracentrifugated FBS containing or not containing 300 ng/mL of myrNef_SF2_w.t were added. After 20 h, media were harvested and underwent differential ultracentrifugations to isolate exosomes (Exo) and microvesicles (MV). (**B**,**D**) Isolated fluorescent vesicles were counted through FC by plotting fluorescence at 525/40 nm (Bodipy C_16_) versus log scale side scatter (SSC-A). The number of exosomes (**B**) or microvesicles (**D**) was registered in the rectangular region, as reported in Materials and Methods. (**C**,**E**) Histograms show the mean total number of fluorescent exosomes and microvesicles normalized to an equal number of cells. Values are expressed as mean ± S.D. of triplicates of three independent experiments. Two-tailed *t* test; **, *p* < 0.01, ns, not significant vs. respective Ctrl. (**F**) Characterization of exosomal markers in exosomes and microvesicles isolated from GEN2.2 cells treated with Nef protein. An equal number of collected vesicles (8 × 10^6^) was loaded on 11% SDS-PAGE gel for Western blot analysis with antibodies specific for Tsg101, Hsp90, Alix, CD81, Flotillin-1 and COXIV and Nef. Western blot analysis of both cell extracts and vesicles was shown (EXO: exosomes; MV: microvesicles). An equal protein content of the cell extracts was loaded, and actin was used as loading control. A representative Western blot, out of three independent experiments that yielded similar results, is shown.

**Table 1 viruses-14-00074-t001:** Cytokines/chemokines released by GEN2.2 cells in response to Nef.

Constitutive Expression	Cytokines/ Chemokines	Fold Increase vs. Ctrl		*p* Value		
		Nef w.t	Nef 4EA	Nef w.t vs. Ctrl	Nef w.t vs. 4EA	4EA vs. Ctrl
**Time 4 h**						
High	MCP-1 (91.47 ± 12.07)	1.54 ± 0.11	0.95 ± 0.12	0.021	0.0169	ns
	TNF-α (773.32 ± 365)	2.07 ± 0.13	1.19 ± 0.01	0.0015	0.0026	ns
	G-CSF (118.98 ± 37.13)	4.23 ± 0.39	1.19 ± 0.09	0.0016	0.0020	ns
	VEGF (139.53 ± 11.56)	2.20 ± 1.15	1.28 ± 0.10	ns	ns	ns
Low	IL-8 (1.44 ± 0.78)	30.40 ± 0.06	1.22 ± 0.32	<0.0001	<0.0001	ns
	IP-10 (3.84 ± 1.32)	5.39 ± 1.98	0.98 ± 0.03	ns	ns	ns
	MIP-1β (10.13 ± 1.14)	4.94 ± 1.96	1.08 ± 0.05	ns	ns	ns
	IL-15 (34.87 ± 35.53)	2.46 ± 0.85	1.02 ± 0.03	ns	ns	ns
**Time 6 h**						
High	MCP-1 (109.83 ± 36.81)	1.35 ± 0.17	0.85 ± 0.01	ns	0.0294	ns
	TNF-α (868.11 ± 447.62)	1.58 ± 0.02	0.95 ± 0.09	0.0035	0.0028	ns
	G-CSF (107.80 ± 28.67)	4.18 ± 2.21	0.86 ± 0.00	ns	ns	ns
	VEGF (163.75 ± 36.01)	1.09 ± 0.14	0.86 ± 0.04	ns	ns	ns
Low	IL-8 (1.77 ± 0.70)	30.23 ± 0.64	1.33 ±0.00	<0.0001	<0.0001	ns
	IP-10 (14.88 ± 3.02)	10.99 ± 6.63	0.99 ± 0.19	ns	ns	ns
	MIP-1β (9.56 ± 3.46)	5.74 ± 3.60	0.81 ±0.14	ns	ns	ns
	IL-15 (10.48 ± 3.22)	3.40 ±3.48	0.95 ± 0.07	ns	ns	ns
**Time 20 h**						
High	MCP-1 (87.92 ± 27.87)	2.83 ± 0.23	1.10 ± 0.04	0.0017	0.0020	ns
	TNF-α (1200 ± 383.55)	1.61 ± 0.25	1.12 ± 0.06	ns	ns	ns
	G-CSF (39.67 ± 23.04)	6.02 ± 2.45	1.16 ± 0.32	ns	ns	ns
	IP-10 (69.37 ± 25.49)	9.05 ± 2.22	1.17 ± 0.00	0.0166	0.0176	ns
	VEGF (573.09 ± 141.74)	1.43 ± 0.80	0.94 ± 0.18	ns	ns	ns
Low	IL-8 (2.14 ± 1.31)	30.80 ± 14.09	2.09 ± 0.86	ns	ns	ns
	MIP-1β (3.61 ± 0.10)	5.13 ± 0.54	1.15 ± 0.07	0.0020	0.0022	ns
	IL-15 (39.02 ± 38.18)	1.44 ± 0.76	0.97 ± 0.07	ns	ns	ns

Footnotes: the concentrations (pg/mL ± SD) detected in the supernatants of controls are reported in brackets. The values are means of two independent experiments evaluated in triplicate. One-way ANOVA and Tukey’s multiple comparisons were used. ns = differences not statistically significant.

**Table 2 viruses-14-00074-t002:** Cytokines/chemokines released by THP-1/PMA cells in response to Nef.

Constitutive Expression	Cytokines/ Chemokines	Fold Increase vs. Ctrl		*p* Value		
		Nef w.t	Nef 4EA	Nef w.t vs. Ctrl	Nef w.t vs. 4EA	4EA vs. Ctrl
**Time 4 h**						
High	IL-8 (1144.8 ± 302.8)	11.00 ± 1.97	1.07 ± 0.24	0.0065	0.0067	ns
	IL-15 (115.2 ± 7.1)	1.83 ± 0.32	1.04 ± 0.20	ns	ns	ns
	G-CSF (3047.1 ± 1297.9)	1.63 ± 0.19	1.21 ± 0.16	0.0441	ns	ns
	MIP-1α (120.81 ± 33.9)	4.00 ± 0.85	1.51 ± 0.57	0.0299	0.0487	ns
	MIP-1β (235.5 ± 60.9)	99.25 ± 0.01	1.37 ± 0.53	<0.0001	<0.0001	ns
	RANTES(1218.7 ± 103.2)	1.60 ± 0.28	0.90 ± 0.00	ns	0.0468	ns
Low	FGF basic (59.7 ± 6.4)	1.89 ± 0.26	1.08 ± 0.11	0.0248	0.0316	ns
	IFN-γ (35.3 ± 21.3)	2.72 ± 0.02	0.89 ± 0.13	0.0005	0.0004	ns
	IP-10 (66.7 ± 27.1)	4.14 ± 0.37	0.96 ± 0.08	0.0015	0.0015	ns
	MCP-1 (49.37 ± 1.7)	5.11 ± 0.16	1.05 ± 0.21	0.0002	0.0002	ns
	TNF-α (34.84 ± 9.8)	257.41 ± 95.33	1.72 ± 0.88	0.0376	0.0379	ns
**Time 6 h**						
High	IL-8 (1409.8 ± 436.8)	30.55 ± 2.90	1.63 ± 0.19	0.0009	0.0009	ns
	IL-15 (126.4 ± 1.65)	1.84 ± 0.08	1.08 ± 0.03	0.0012	0.0015	ns
	G-CSF (3368.5 ± 1157.7)	1.64 ± 0.23	1.27 ± 0.10	0.0428	ns	ns
	MIP-1α (156.8 ± 31.2)	3.13 ± 0.04	1.97 ± 0.67	0.0242	ns	ns
	MIP-1β (296.7 ± 39.7)	84.60 ± 0.85	1.67 ± 0.39	<0.0001	<0.0001	ns
	RANTES (1644.3 ± 221.3)	2.26 ± 0.09	1.01 ± 0.16	0.0028	0.0028	ns
Low	FGF basic (64.8 ± 9.4)	1.92 ± 0.02	1.11 ± 0.02	<0.0001	<0.0001	0.0142
	IFN-γ (42.8 ± 25.8)	3.05 ± 0.36	1.06 ± 0.09	0.0048	0.0052	ns
	IP-10 (84.2 ± 15.1)	15.19 ± 0.44	1.45 ± 0.49	<0.0001	<0.0001	ns
	MCP-1 (70.3 ±6.3)	9.55 ± 0.36	1.33 ± 0.18	<0.0001	<0.0001	ns
	TNF-α (40.4 ± 19.1)	247.33 ± 18.29	2.11 ± 0.58	0.0004	0.0004	ns
**Time 20 h**						
High	IL-8 (1926.2 ± 37.0)	65.46 ± 72.21	31.21 ± 36.79	ns	ns	ns
	IL-15 (112.7 ± 10.6)	2.18 ± 0.12	1.63 ± 0.46	0.0476	ns	ns
	G-CSF (2778.9 ± 1397.5)	2.05 ± 0.07	1.95 ± 0.22	0.0086	ns	0.011
	MIP-1α (125.8 ± 56.2)	4.18 ± 0.96	4.25 ± 1.34	ns	ns	ns
	MIP-1β (483.1 ± 160.3)	47.01 ± 0.86	13.15 ± 15.77	0.0297	ns	ns
	RANTES (4473.3 ± 1893.8)	4.82 ± 1.80	2.26 ± 1.08	ns	ns	ns
Low	FGF basic (55.6 ± 0.86)	2.33 ± 0.47	1.78 ± 0.68	ns	ns	ns
	IFN-γ (60.8 ± 31.3)	4.21 ± 0.15	2.37 ± 1.37	ns	ns	ns
	IP-10 (368.4 ± 168.3)	25.52 ± 1.16	3.43 ± 1.60	0.0005	0.0007	ns
	MCP-1 (176.4 ± 27.7)	16.74 ± 0.48	10.16 ± 11.40	ns	ns	ns
	TNF-α (51.4 ± 15.9)	123.00 ± 21.50	11.06 ± 12.54	0.0070	0.0090	ns

Footnotes: the concentrations (pg/mL ± SD) detected in the supernatants of controls are reported in brackets. The values are means of two independent experiments evaluated in triplicate. One-way ANOVA and Tukey’s multiple comparisons were used. ns = differences not statistically significant.

## Data Availability

Data sharing is not applicable to this article.

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
