# Peer review of "HIV-1 Nef Protein Affects Cytokine and Extracellular Vesicles Production in the GEN2.2 Plasmacytoid Dendritic Cell Line"

_viruses, 2021, doi:10.3390/v14010074_

Round 1

Reviewer 1 Report

In this manuscript Aiello A. and colleagues reveal a new biological function of myristylated HIV-1 Nef protein showing it activates the plasmacytoid dendritic cell line GEN2.2. Added exogenously to these cells, which are used as model-pDCs as they share common features with primary plasmacytoid dendritic cells, myrNEF induces cytokine production ( MCP1, TF-alpha, G-CSF, IL-8, IP-10, MIP1-beta) which in turn activates Stat-1 and Stat-2 and influence gene expression program (Stat1, IRF1, ISG15) suggesting IFN-I/IFN-L production (although not detectable directly). They also show that myrNef enters pDCs , do not increase EV release. It is indeed  associated with exosomes (small size vesicles) ans reduces their production.

Although GEN2.2 cells may not behave as natural PDCs, these data suggest that Nef may contribute to activate pDCs in vivo, and sustain chronic immune activation in HIV infected patients, which is associated with HIV pathogenesis. This observation suggests a potential new mechanism for pDC activation in HIV infection in addition to the known RNA/TLR7 dependent activation induced by HIV particles (dead or alive) and HIV infected cells.

The manuscript is well written, cautiously detailed in terms of methods with rare omissions, and conclusions in adequation with observations.

It deserves the following comments:

Major comments :

  • The use of a cell line instead of primary pDCS in this work is comprehensive for technical reasons, but deserves some comments. The sentence in results page 7 line 347-349 outlining common features between primary pDCs and GEN2.2 cell line could also refer to Carmona-Saez P. et al (Bioinformatics, 33(23), 2017, 3691–3695, doi: 10.1093/bioinformatics/btx502 ) in addition to ref 38 to support the claim of the authors that GEN2.2 cell line shares features with primary pDCs. This reference indeed extends these common features by the comparison of primary pDCs to GEN2.2 cell line at the gene signature level. Nevertheless, the authors should also be more critic on the limits of the use of this cell line in their discussion because of the same reference. Indeed this paper also clearly comment that « Also of note, pathways related with transcription regulation, signal transduction and intracellular traffic were consistently down-regulated in GEN2.2 and CAL-1 compared with pDC (see additional file 4). » and « some pathways were under-represented in CAL-1 and GEN2.2, such as negative regulation of T cell differentiation and collagen catabolic processes. » which suggest that gene expression levels and pathways are somewhat different in this cell line compared to primary pDC. Preliminary data with primary pDCs would therefore be a plus to this manuscript, and, at least, authors should take this comment into account by revising the discussion.
  • In line with my precedent comment, in the title of the manuscript  « GEN2.2 plasmacytoid dendritic cells » should be replaced by « GEN2.2 plasmacytoid dendritic cell line » to avoid any confusion.
  • The author show in Figure 1 that myrNef internalizes in GEN2.2 cells but they did not show similar data for the mutant NEF which is used in the manuscript as a negative control and control that Tyrosine phosphorylation by Nef is implicated in the activation mechanism. These data should be included in Figure 1 to be shure that mutant NEF does also enter the cells and that the absence of phosphorylation of STAT 1 and 2 in mutant NEF control is not because of absence of internalization.

  • In the discussion page 22 line 753-755 suggesting the induced cytokine production could favor recruitment and activation of CD4+ T cells, the authors should discuss and refer to Li et al who indeed provided evidences in the SIV/macaque model that vaginal exposure to SIV induces local mucosal pDC activation and their production of IFN alpha and chemokines (Mip1-alpha and beta) very early on, which is followed by CCR5+CD4 T cell recruitment a mechanism by which pDC could fuel SIV replication (Li Q. et al Nature. 2009 April 23; 458(7241): 1034–1038. doi:10.1038/nature07831).

Minor comments :

  • Figure 2C is cited in the text before Figure 2B (Page 9 and page 10 respectively) which is unconventional.
  • In methods, paragraph 2.3 page, the clones or commercial references for antibodies used should be added.
  • I was a little bit confused with figure 2 at first reading because 2 different kinetics (shrt vs long term) are used in Figure A versus B and in C versus D which in not clearly introduced in the text page 9-10 and not stated in figure 2 legend. Please clarify.

Author Response

Point 1: The use of a cell line instead of primary pDCS in this work is comprehensive for technical reasons, but deserves some comments. The sentence in results page 7 line 347-349 outlining common features between primary pDCs and GEN2.2 cell line could also refer to Carmona-Saez P. et al (Bioinformatics, 33(23), 2017, 3691–3695, doi: 10.1093/bioinformatics/btx502 ) in addition to ref 38 to support the claim of the authors that GEN2.2 cell line shares features with primary pDCs. This reference indeed extends these common features by the comparison of primary pDCs to GEN2.2 cell line at the gene signature level. Nevertheless, the authors should also be more critic on the limits of the use of this cell line in their discussion because of the same reference. Indeed this paper also clearly comment that « Also of note, pathways related with transcription regulation, signal transduction and intracellular traffic were consistently down-regulated in GEN2.2 and CAL-1 compared with pDC (see additional file 4). » and « some pathways were under-represented in CAL-1 and GEN2.2, such as negative regulation of T cell differentiation and collagen catabolic processes. » which suggest that gene expression levels and pathways are somewhat different in this cell line compared to primary pDC. Preliminary data with primary pDCs would therefore be a plus to this manuscript, and, at least, authors should take this comment into account by revising the discussion.

Response 1: Thank you for the important comment and suggestion. We have added the indicated reference (line 541), inserted a comment in the discussion regarding the difference between GEN2.2 and pDCs (lines 1029-1038) and added preliminary experiments on pDCs at the beginning of the manuscript (pages 9-12).

Point 2: In line with my precedent comment, in the title of the manuscript  « GEN2.2 plasmacytoid dendritic cells » should be replaced by « GEN2.2 plasmacytoid dendritic cell line » to avoid any confusion.

Response 2: We have changed the title with your proposed alternative: “HIV-1 Nef protein affects cytokine and extracellular vesicles production in the GEN2.2 plasmacytoid dendritic cell line”.

Point 3: The authors show in Figure 1 that myrNef internalizes in GEN2.2 cells but they did not show similar data for the mutant NEF which is used in the manuscript as a negative control and control that Tyrosine phosphorylation by Nef is implicated in the activation mechanism. These data should be included in Figure 1 to be sure that mutant NEF does also enter the cells and that the absence of phosphorylation of STAT 1 and 2 in mutant NEF control is not because of absence of internalization.

Response 3: Thank you for the suggestion. We have inserted the image corresponding to the internalization of the 4EA mutant (see Figure 4). As shown by the confocal images and confirmed by western blot analysis, Nef4EA is incorporated in the cells like the wild type protein.

Point 4: In the discussion page 22 line 753-755 suggesting the induced cytokine production could favor recruitment and activation of CD4+ T cells, the authors should discuss and refer to Li et al who indeed provided evidences in the SIV/macaque model that vaginal exposure to SIV induces local mucosal pDC activation and their production of IFN alpha and chemokines (Mip1-alpha and beta) very early on, which is followed by CCR5+CD4 T cell recruitment a mechanism by which pDC could fuel SIV replication (Li Q. et al Nature. 2009 April 23; 458(7241): 1034–1038. doi:10.1038/nature07831).

Response 4: The indicated reference has been added and discussed in light of the results reported for the cytokine production (lines 1029-1038).

Minor comments :

Point 1: Figure 2C is cited in the text before Figure 2B (Page 9 and page 10 respectively) which is unconventional.

Response 1: We have modified the sequence of the figures in the text as suggested (page 16, lines 628-637).

Point 2: In methods, paragraph 2.3 page, the clones or commercial references for antibodies used should be added.

Response 2: We have added the missing information (lines 198-205).

Point 3: I was a little bit confused with figure 2 at first reading because 2 different kinetics (shrt vs long term) are used in Figure A versus B and in C versus D which in not clearly introduced in the text page 9-10 and not stated in figure 2 legend. Please clarify.

Response 3: Thank you for your comment. We have better clarified the topic in the text (page 16, lines 627-629) and in the corresponding legend of Figure 5.

Reviewer 2 Report

This paper is interesting but it is written in a style that should be improved. Please correct the sentences on l. 430,('"what observed"), and l. 648, 684, 732, and 793 ('"what reported"): the verb ('was") is missing every time! 

Also please correct l.407 and l.530: "show that.."(not: "show as"); l.490: "to what extent (not "in");  l.821: "considering the above" (not:"considered"); l.l.573; "as a result" (not "as result"); l.740: "because it required" (not: "because required"); etc...

The use of initials with no explanation is rather difficult to cope with: what is for instance the meaning of" FTCC"? "APC"?"FITC"?or "PE"?

The presentation of the Western blot in Fig 3 is totally incomprehensible: the 90 kDa band is repeated twice! The 48 kDa band is presented below the 34 kDa one! And then there is a Nef 20µg band in the air!!   

Fig 4 is redundant with Table 1 and not very clear.

Author Response

RESPONSE TO REVIEWER 2 COMMENTS

Point 1: This paper is interesting but it is written in a style that should be improved. Please correct the sentences on l. 430,('"what observed"), and l. 648, 684, 732, and 793 ('"what reported"): the verb ('was") is missing every time! 

Response 1: We thank the reviewer for the suggested edits. We made the corrections as suggested.

Point 2: Also please correct l.407 and l.530: "show that.."(not: "show as"); l.490: "to what extent (not "in");  l.821: "considering the above" (not:"considered"); l.l.573; "as a result" (not "as result"); l.740: "because it required" (not: "because required"); etc...

Response 2: We made the corrections as suggested.

Point 3: The use of initials with no explanation is rather difficult to cope with: what is for instance the meaning of" FTCC"? "APC"?"FITC"?or "PE"?

Response 3: We provided the complete explanation for the acronyms.

Point 4: The presentation of the Western blot in Fig 3 is totally incomprehensible: the 90 kDa band is repeated twice! The 48 kDa band is presented below the 34 kDa one! And then there is a Nef 20µg band in the air!!   

Response 4: In the western blot of Figure 1 (now Figure 4), incorrectly indicated as Figure 3 in the comment, the 90 kDa band was repeated twice because the phosphorylated form of STAT1 protein and the not phosphorylated protein are at the same height on the membrane. For this reason, the nitrocellulose membrane was first incubated with anti-P-STAT1 antibody, then stripped and incubated with anti-STAT1 antibody. The 48kDa band was moved above that of 34kDa. The band of 20ng of Nef is not in the air. Nef protein was loaded separately from the others samples by leaving an empty well in between to avoid signal interference with neighbouring bands.

Point 5: Fig 4 is redundant with Table 1 and not very clear.

Response 5: We added Figure 4 because it could help to understand at first glance the marked differences in the pool of cyto/chemokines released by the two cell lines. Table 1 refers only to the cyto/chemokine that are mainly modulated. We prefer to maintain both table and figure.

Reviewer 3 Report

The manuscript by Aiello et al describes the effects of recombinant Nef on GEN2.2 plasmacytoid dendritic cell line. They show that Nef activates STAT signaling and induces production of pro-inflammatory cytokines. Nef stimulated release of exosomes, but not microvesicles, and was itself incorporated into exosomes. While these observations are potentially interesting, they do not seem to assemble into a story, and their relevance to HIV disease is unclear. The reasons are listed below.

  1. The authors investigate the effects of recombinant Nef, whereas most recent studies indicate that Nef is released from infected cells predominantly as a component of EVs.
  2. It remains unclear how Nef enters the cells and initiates signaling. Lack of activity of the mutant Nef may provide some clues, but the authors did not pursue this line of investigations. Without understanding the initiating events, their analysis of STAT activation does not have much value.
  3. The authors show incorporation of Nef into exosomes, but it is unclear whether this event has any biological implications.
  4. Although GEN2.2 cells may mimic certain properties of pDC, experiments with primary cells are needed to prove relevance of the findings.

Author Response

RESPONSE TO REVIEWER 3 COMMENTS

Point 1: The authors investigate the effects of recombinant Nef, whereas most recent studies indicate that Nef is released from infected cells predominantly as a component of EVs.

Response 1: Although Nef is released from infected cells predominantly as a component of EVs or transferred by nanotubes, Nef protein can be released by necrotic cells into the extracellular space and be internalized by neighbouring cells.

Moreover, the aim of this study was also to investigate the effects of the Nef protein on pDCs as a possible component of a vaccine. 

Point 2: It remains unclear how Nef enters the cells and initiates signaling. Lack of activity of the mutant Nef may provide some clues, but the authors did not pursue this line of investigations. Without understanding the initiating events, their analysis of STAT activation does not have much value.

Response 2: We agree with your comment. Regarding the entry of the Nef protein into the cells, some experiments were conducted by testing inhibitors of different pathways such as cytochalasin D, imipramine, and LY294002 that inhibit the actin polymerization, macropinocytosis and PI3K pathway, respectively. However, none of these inhibitors affected the entry of the protein, suggesting that other mechanisms must be involved. For instance, Nef protein is characterized by the presence of a myristic acid and some specific aa residues at the N-terminal arm that anchor the protein to artificial membranes and favour interaction with the cell membranes. In this regard, the mutant G2A, lacking the myristic acid and tested by our laboratory, showed a lower level of internalization and lack the ability to induce STATs activation, suggesting the importance of this domain in the entry process (see also Gerlach H, Laumann V, Martens S, Becker CF, Goody RS, Geyer M. HIV-1 Nef membrane association depends on charge, curvature, composition and sequence. Nat Chem Biol. 2010 Jan;6(1):46-53). We have previously published that myrNef treatment of primary macrophages or THP-1 differentiated cells induces the secretion of inflammatory factors and IFN beta (Mangino et al., J.Virol.,2007), and both the myristoylation and the conserved 4E acidic cluster of the protein are required to induce this effect. Through silencing procedure and co-immunoprecipitation experiments performed in  THP-1 cell line, we have published that Nef-induced effects require also specific intracellular adaptors (Tumor necrosis factor receptor–associated factor TRAF2 and 6) (Mangino G, Percario ZA, Fiorucci G, Vaccari G, Acconcia F, Chiarabelli C, Leone S, Noto A, Horenkamp FA, Manrique S, Romeo G, Polticelli F, Geyer M, Affabris E. HIV-1 Nef induces proinflammatory state in macrophages through its acidic cluster domain: involvement of TNF alpha receptor associated factor 2. PLoS One. 2011;6(8):e22982).

We added some speculation at the end of the discussion to prompt the interest in the field (page 31).

Point 3: The authors show incorporation of Nef into exosomes, but it is unclear whether this event has any biological implications.

Response 3: The biological relevance of the incorporation of Nef into exosomes has been reported by different works in literature, that were cited in the introduction (lines 82-86). For instance, Nef-containing EVs induce T-cell apoptosis (Lenassi, M et al., Traffic Cph. Den. 2010), make resting CD4+ T lymphocytes competent for HIV expression and replication, reactivate cells latently infected with HIV-1 (Arenaccio et al., J. Virol. 2014; Arenaccio et al., Retrovirology 2014; Arenaccio et al., Retrovirology 2015; Ostalecki et al., EBioMedicine 2016), as well as enhance the levels of cytokines and chemokines such as IL-2, IL-8, IL-6, RANTES and IL-17A (Raymond et al., J. Neurovirol. 2016). However, the analysis of the Nef containing-exosome fraction described in the paper and its biological activity requires more in depth investigation that goes beyond the aim of the present work.

Point 4: Although GEN2.2 cells may mimic certain properties of pDC, experiments with primary cells are needed to prove relevance of the findings.

Response 4: We thank the Reviewer for raising this point that did help to improve the manuscript. We have inserted in the paper experiments performed on PBMC depleted or not of pDCs and on primary pDCs that suggested us an effect of Nef on this primary dendritic subset and prompted us to work with GEN2.2 cells being aware of the limitations of the model.  

Round 2

Reviewer 1 Report

The authors have appropriately addressed my comments. The manuscript is now strongly improved thanks to their work.

I still have some interrogation in concern with the internalization of the mutant Nef. The confocal images show indeed that the mutant protein is internalized, but it is difficult to ascertain from these images that the amount of protein internalized is equivalent, wt versus mutant protein. If these images can be quantified or data showing that the % of cells that do internalize mutant Nef is equivalent to those internalizing wt Nef, this information would be more convincing. A graph could then be added to the figure 4A with statistical analysis to certify the equivalence.

One typo should be corrected: line 616 the word "that" is doubled, please correct typo

Author Response

RESPONSE TO REVIEWER 1 COMMENTS

Point 1. I still have some interrogation in concern with the internalization of the mutant Nef. The confocal images show indeed that the mutant protein is internalized, but it is difficult to ascertain from these images that the amount of protein internalized is equivalent, wt versus mutant protein. If these images can be quantified or data showing that the % of cells that do internalize mutant Nef is equivalent to those internalizing wt Nef, this information would be more convincing. A graph could then be added to the figure 4A with statistical analysis to certify the equivalence.

Response 1: Thank you for the suggestion. We agree with the difficulty to ascertain from the confocal images that the amount of protein internalized is equivalent. For this reason, we reported also a western blot showing the comparable amount of the wild type and 4EA protein internalized at both 4 and 24 hours in GEN2.2 cells. To further strengthen this information, we have added a graph comparing the % of GEN2.2 cells that internalize the two proteins, as required.

Point 2. One typo should be corrected: line 616 the word "that" is doubled, please correct typo.

Response 2: Thank you for the correction. We have delated the double “that”.

Reviewer 3 Report

The revised version provides much needed improvements. However, several concerns remain.

  1. Although the Introduction now provides some rationale for studying the effects of free Nef, the authors should clearly  acknowledge that the Nef associated with exosomes is the predominant form of Nef in circulation, and the effects described in the manuscript may differ from the effects of Nef EVs. In this regard, it may be helpful to discuss the results considering literature devoted to the effects of Nef EVs.
  2. Fig. 1: The authors now show results with primary cells. Given variability between the donors, these results should be presented not only for individual donors (even representative), but also as a graph showing all donors (3 or more) with corresponding statistics.
  3. Line 558: 'pDCs are less sensitive to Nef treatment with respect to primary macrophages'. It is unclear where this comes from, as macrophages are not analyzed in this experiment, and GEN2.2 cells are not real pDCs.

Author Response

RESPONSE TO REVIEWER 3 COMMENTS

Point 1. Although the Introduction now provides some rationale for studying the effects of free Nef, the authors should clearly  acknowledge that the Nef associated with exosomes is the predominant form of Nef in circulation, and the effects described in the manuscript may differ from the effects of Nef EVs. In this regard, it may be helpful to discuss the results considering literature devoted to the effects of Nef EVs.

Response 1: Thank you for the suggestion. We improved the discussion accordingly.

Point 2. Fig. 1: The authors now show results with primary cells. Given variability between the donors, these results should be presented not only for individual donors (even representative), but also as a graph showing all donors (3 or more) with corresponding statistics.

Response 2: Thank you for your comment. We added the graphs showing the corresponding statistics.

Point 3. Line 558: 'pDCs are less sensitive to Nef treatment with respect to primary macrophages'. It is unclear where this comes from, as macrophages are not analyzed in this experiment, and GEN2.2 cells are not real pDCs.

Response 3. Thank you for the comment. We corrected the sentence as follows by better clarifying the concept:

Considering these results, we can infer that GEN2.2 cells are less sensitive to Nef treatment with respect to what we previously observed in primary macrophages. In particular, in primary macrophages STAT1 tyrosine phosphorylation is induced by the release of cytokines and chemokines with lower concentrations of the viral protein (10-100 ng/ml) and earlier, i.e., after only 2h of cell treatment with Nef (Mangino et al., 2007; Federico et al., 2001).
